# Undersizing of Aged African Biomass Burning Aerosol by an Ultra High Sensitivity Aerosol Spectrometer

Steven G. Howell[1], Steffen Freitag[1], Amie Dobracki[1,2], Nikolai Smirnow[1,3], and Arthur J. Sedlacek III[4]

[1]University of Hawaii at Manoa, Department of Oceanography, Honolulu, HI USA
[2]Now at Rosenstiel School of Marine and Atmospheric Science, University of Miami, Miami, FL USA
[3]Now at NetApp, Boulder, CO USA
[4]Brookhaven National Laboratory, US Department of Energy, Upton, NY USA

**Correspondence:** Steven Howell (sghowell@hawaii.edu)

**Abstract.** The Ultra-High Sensitivity Aerosol Spectrometer (UHSAS) differs from most other optical particle spectrometers by using a high-power infrared (IR) laser to detect small particles and reduce the sizing ambiguity due to the non-monotonicity of scattering with particle size.

During the NASA ORACLES project (ObseRvations of Aerosols above CLouds and their intEractionS) over the southeast Atlantic Ocean, the UHSAS clearly undersized particles in the biomass burning plume extending from Southern Africa. Since the horizontal and vertical extent of the plume was vast, the NASA P-3B research aircraft often flew through a fairly uniform biomass burning plume for periods exceeding 30 minutes, sufficient time to explore the details of the UHSAS response by selecting single particle sizes with a Differential Mobility Analyzer (DMA) and passing them to the UHSAS. This was essentially an in-flight calibration of the UHSAS using the particles of interest. Two modes of responses appeared. Most particles were undersized by moderate amounts, ranging from not at all for 70 nm aerosols to 15 % for 280 nm particles. Mie scattering calculations show that composition-dependent refractive index of the particles cannot explain the pattern. Heating of brown carbon or tarballs in the beam causing evaporation and shrinking of the particles is the most plausible explanation, though missizing to to non-sphericity cannot be ruled out. 10–30 % of the particles were undersized by 25 to 35 %. Those were apparently the particles containing refractory black carbon. Laboratory calibrations confirm that black carbon is drastically undersized by the UHSAS, because particles heat to their vaporization point and shrink.

A simple empirical correction equation was implemented that dramatically improves agreement with DMA distributions between 100 and 500 nm. It raised median particle diameter 18 nm, from 163 to 181 nm during the August 2017 deployment and by smaller amounts during deployments with less intense pollution. Calculated scattering from UHSAS size distributions increased by about 130 %, dramatically improving agreement with scattering measured by nephelometers. The correction is only valid in polluted instances; clean marine boundary layer and free troposphere aerosols behaved more like the calibration spheres. We were unable to directly test the correction between 500 and 1000 nm, though APS data appear to show that the correction is poor at the largest diameters, which is no surprise as the composition of those particles is likely to be quite different than that of the accumulation mode. This adds to the evidence that UHSAS data must be treated cautiously whenever the aerosol may absorb infrared light. Similar corrections may be required whenever brown carbon aerosol is present.

## 1 Introduction

Particles in the air, or aerosols, play a major role in the atmosphere. They directly affect the radiation balance of the Earth by scattering sunlight, and indirectly by affecting cloud properties (Twomey, 1977; Albrecht, 1989; Boucher et al., 2013). They participate in geochemical cycling of nutrients (e.g., Chadwick et al., 1999) and pollutants, and can have negative impacts on
human health (e.g., Woodcock, 1948; Shiraiwa et al., 2017; Burnett et al., 2018).

The wide span of aerosol sizes (from $< 1\,\text{nm}$ to $> 100\,\text{µm}$), composition, and shapes mean there is no wholly satisfactory method of measuring particle size. In the case of non-spherical particles, there is not even a single definition of diameter that is universally applicable (Baron and Willeke, 1993).

Optical particle counters (OPCs) that work by measuring individual particle scattering from a light beam are appealing
because they can have high size resolution and rapidly measure particles in the diameter ranges that most affect health, radiation, and cloud properties. A wide variety of these instruments have been used in field projects over the last few decades (e.g., Whitby and Vomela, 1967; Clarke, 1991; Gebhart, 1993; Ames et al., 2000; Hand and Kreidenweis, 2002; Haywood et al., 2003; McNaughton et al., 2009; Rosenberg et al., 2012). The main drawback is that the amount of scattered light detected by an OPC is affected by particle shape, composition, inhomogeneities, wavelength of light, and the angles between particle, light
beam, and the detection optics. To make matters worse, even in the ideal case, with homogeneous spheres of known refractive index, resonances between the light waves and the particles mean that the amount of light scattered is non-monotonic with diameter for particles near the wavelength of light.

There are a few ways to mitigate these resonances, sometimes called "Mie wiggles", after Gustav Mie, who first solved the problem of light scattering due to arbitrary diameter homogeneous spheres (Mie, 1908). One can use white light (Chen
et al., 1984; Liu et al., 1985); concentrate on forward scattering, where Mie wiggles are reduced (Gebhart, 1991); do inversions to rationally assign particles with a given scattering to appropriate sizes (Ames et al., 2000; Wang, 2002; Rosenberg et al., 2012); smooth the response curve (e.g., Robinson and Lamb, 1986; Clarke, 1991) or broaden the size bins to accommodate the uncertainty.

This uncertainty is particularly pronounced when trying to calculate higher moments, such as surface area or particle mass.
Since those vary with the square and cube of diameter, small errors in sizing are magnified considerably. One can partially compensate by determining how scattering would be affected by a refractive index calculated from particle composition, but the fundamental ambiguity remains.

### 1.1 The UHSAS

Droplet Measurement Technology's Ultra-High Sensitivity Aerosol Spectrometer (UHSAS) (Cai et al., 2008; Kupc et al.,
2018) approaches this problem by using infrared (IR) light ($1054\,\text{nm}$), keeping Mie scattering monotonic through the UHSAS maximum diameter of $1000\,\text{nm}$ and meaning particles below $300\,\text{nm}$ are within the Rayleigh regime, where scattering goes

with the 6th power of diameter, giving a nice log-linear relationship over much of the sizing range. This has the additional benefit of suppressing the effect of shape: when the particle size is much smaller than wavelength, scattering is determined primarily by the volume of the particle rather than the cross section (Gebhart, 1993). It requires an intense laser to detect the smallest particles ($\sim 1\,\mathrm{kW}$ circulating in the sample volume) and wide-ranging amplification to handle the $> 6$ orders of magnitude scattering change over the 60 to 1000 nm nominal detection limits. To achieve this, the UHSAS uses a pair of detectors, one an avalanche photodiode and the other a low gain PIN photodiode. Each photodiode has two output channels with different gains, yielding 4 channels spanning the range of particle scattering.

The UHSAS sample flow rate can vary from 1 to $100\,\mathrm{std\,cm^3\,min^{-3}}$ (referenced to 298 K and 1013 hPa), and is aero-dynamically focused with a nozzle of 500 μm diameter and a sheath flow typically about $700\,\mathrm{cm^3\,min^{-3}}$ through a 760 μm nozzle positioned within 1 mm of the laser. This jet creates a particle beam roughly 50 to 100 μm in diameter, much smaller than the 500 μm $1/e^2$ intensity diameter of the laser (Droplet Measurement Technologies, 2013).

As with any optical particle sizer, the UHSAS is subject to sizing errors when the refractive index of sample particles is different than that of the calibration material. One can approach that by using calibration materials close to the refractive index of natural aerosol (Sawamura et al., 2017), use Mie scattering calculations and composition to correct calibrations for sampled aerosol (Ames et al., 2000; Cai et al., 2008), or conclude that the errors are small enough to ignore (Volkamer et al., 2015). But it appears that the UHSAS may have unique problems with black carbon. Yokelson et al. (2011) noted poor behavior of the UHSAS in a Mexican biomass burning (BB) plume and concluded that strongly absorbing particles are essentially invisible to the UHSAS.

## 1.2 A note about black carbon

Light absorbing carbonaceous material (LAC) is one of the most difficult parts of aerosol to describe chemically, to measure, and to model (Bond and Bergstrom, 2006). It is poorly defined chemically, tends to be highly aspherical, and is present in highly variable quantities. The plethora of names invented to describe it gives an indication of the complexity: soot, elemental carbon (EC), black carbon (BC), refractory black carbon (rBC), brown carbon (BrC), and tarballs all address subtly different properties. Historically, it has been practical to measure either the amount of carbon with chemical methods or the blackness with absorption measurements, but quantitatively connecting the two amidst the complexity of ambient aerosol remains a challenge (Petzold et al., 2013).

A review of this is beyond the scope of this paper (see Bond and Bergstrom 2006; Petzold et al. 2013; Michelsen et al. 2020), but a description of how the terms are used in this paper may be useful. BC is an umbrella term for carbon that absorbs strongly across spectrum from infrared through ultraviolet with a fairly weak ($\sim 1/\lambda$) wavelength dependence. The carbon atoms largely sp$^2$-bonded like graphite. It is refractory and insoluble in water and organic solvents (Bond and Bergstrom, 2006). rBC refers to the LAC particles detected by a DMT Single Particle Soot Photometer (SP2) or other laser-induced incandescence (LII) instrument, which heats IR-absorbing particles to vaporization temperatures, roughly 4000 K for graphite, where it is detected by incandescence (Petzold et al., 2013). That clearly includes BC, but other carbonaceous species can masquerade as BC if they char as they heat (Sedlacek et al., 2018b). BrC is organic material that absorbs light primarily at short wavelengths (Andreae

and Gelencser, 2006) and to a small and very poorly known extent in the IR (e.g. Li et al., 2020; Sumlin et al., 2018b). Tarballs (Pósfai et al., 2003, 2004) are a distinctively spherical variety of BrC prevalent in aged biomass burning plumes. Their optical properties are also poorly known, though it appears that they may absorb more in the IR than other BrC (Alexander et al., 2008; Hoffer et al., 2017; Sedlacek et al., 2018a).

The shape of LAC particles poses another set of difficulties. Electron microscopy of particles freshly emitted from fires shows that soot particles are aggregations of tiny spherules with diameter $\leq 50$ nm tacked together in branching, almost fractal structures. As those particles age, they collapse into more compact shapes, though they typically remain far from spherical. This non-sphericity affects aerodynamic drag, so mobility measurements overestimate particle size (compared to mass or volume-equivalent diameter) and changes optical properties (scattering and absorption) in ways that are not straightforward to
determine (e.g Mackowski, 2006; Sorensen, 2011; Sorensen et al., 2018).

### 1.3   ORACLES (ObseRvations of Aerosols above CLouds and their intEractionS)

ORACLES was a NASA-funded project to examine the direct, indirect, and semi-direct influence of aerosol from burning fields in Africa on the radiative balance over the southeast Atlantic Ocean (Redemann et al., 2020). It was an aircraft-based project with field deployments in September 2016, August 2017, and October 2018. The NASA P-3B was deployed each year, with
an extensive payload of aerosol, cloud droplet, radiation, and remote sensors. See https://espo.nasa.gov/oracles for a project description and links to data and other publications.

The Hawaii Group for Environmental Aerosol Research deployed a UHSAS in addition to a set of Differential Mobility Analyzers (DMAs) to take advantage of the superior time resolution of the UHSAS (1 s or less compared with 90 s), particularly valuable during vertical profiles and when sampling in and around clouds. The UHSAS also offered detection limits of 60 nm,
sufficient to detect nearly all particles likely to activate within clouds.

Aerosol in the ORACLES project was largely aged smoke from burning fields and forests in southern Africa. Plume ages were typically 2 days to 2 weeks, and the aerosol was dominated by organic material with substantial rBC. Data from the first two years showed that when compared to size distributions from the DMAs the UHSAS consistently sized particles too small.

During the final year, we installed tubing and valves to allow a UHSAS to sample size-selected particles from a DMA during
flight. This allowed us to directly test any sizing anomalies due to the characteristics of the ambient aerosol. It was essentially an in-flight UHSAS calibration using particles representative of the plume. This kind of calibration while sampling is not new, as Stolzenburg et al. (1998) did essentially the same thing, but it is the first aircraft deployment we are aware of in a biomass burning plume. We also anticipated that by measuring scattering from aerosol of known size, we could determine refractive index as was done by Hand and Kreidenweis (2002), but on a single-particle basis.
Instead, as will be shown below, we got a dataset from the UHSAS that did not agree with scattering measurements or other sizing techniques, so in this work we explore the reasons for the poor sizing performance of the UHSAS and attempt to make the data as useful as possible by developing a correction scheme.

This test was only possible because the plume we were studying was vast (Pistone et al., 2019; Redemann et al., 2020; Shinozuka et al., 2020); we often spent a half hour or more in remarkably constant aerosol so there was plenty of time to try

time-consuming procedures. This is normally impractical with fast-moving aircraft ($\sim 100\,\mathrm{m\,s^{-1}}$) even considerably downwind of a fire.

## 2 Methods

In the lab and in the field, all sizing instruments were calibrated with polystyrene latex (PSL) spheres (real refractive index $n = 1.572$ at the UHSAS laser wavelength) from 70 to 800 nm diameter. DMA high voltage amplifiers and flow rates were
checked regularly between PSL calibrations. The UHSAS maintained a sample flow rate of $50\,\mathrm{std\,cm^3\,min^{-1}}$ and a sheath flow of $460\,\mathrm{cm^3\,min^{-1}}$. Since the particle speed is determined primarily by the sheath flow, which is volumetrically controlled, particle time in the laser beam did not change much with altitude. Because the P-3B does not fly nearly as high as the NASA DC-8, we did not change the sample flow controller from mass flow to volumetric flow, as was done by Kupc et al. (2018).

### 2.1 Lab Tests

In addition to the field deployment, we tested various materials in the laboratory, partly to familiarize ourselves with the UHSAS, but also to explore the sizing problems evident in the first 2 years. It was not practical to generate good proxies for the aged BB particles present in the field, so we tested some representative non-absorbing salts ($NaCl$, $Na_2SO_4$, and $H_2SO_4$) and two strongly-absorbing materials containing refractory black carbon (rBC): Aquadag, a suspension of graphite flakes, and fullerene soot, which is aggregations of 1—100 nm spherules (Moteki et al., 2009) which are $\sim 90\,\%$ amorphous black carbon[1]
and $\sim 10\,\%$ fullerenes, chiefly the $C_{60}$ form (Gysel et al., 2011). The fullerene soot was from Alfa Aesar (stock #40971, lot #FS12S011), the same lot as used by Gysel et al. (2011).

Aerosol materials were tested by nebulizing aqueous solutions of the material of interest, mixing in dry air to reduce relative humidity (RH) below 30 % to evaporate water from the particles, passing the particles through a long DMA (LDMA), and then into the UHSAS. The LDMA was a modified TSI 3071A; essentially only the DMA column remains; flow control, neutralizers,
high-voltage amplifiers, and software have all been replaced. The LDMA sheath air was desiccated, so particles were selected at $< 5\,\%$ RH. For the fullerene and Aquadag tests, sample air was heated to 450°C in a tube furnace to remove any volatile material before entering the DMA. Gysel et al. (2011) tested fullerene soot and Aquadag aerosol passing through a thermal denuder operating at 400°C and found that it effectively removed organic carbon, leaving particles that were $\sim 87\,\%$ rBC by mass.

DMAs classify particles by the balance between electrostatic attraction and air resistance, so singly charged particles of a given size emerge simultaneously with doubly or even triply charged particles that have twice or thrice the drag. While this is a complication when inverting DMA data into size distributions, it is useful when calibrating an OPC, as the multiply charged particles show up as separate peaks in the distribution. This has the effect of extending the calibrations to diameters that are

---

[1]"Amorphous carbon" in this context is not the definition recommended by Michelsen et al. (2020) which refers to relatively small molecules in incipient or young soot; it is probably more like the turbostatic or polycrystalline graphite where graphitic regions are very small.

in a sense greater than the DMA can select. In the lab, triply charged peaks were sometimes distinct enough to be useful, and double charges were sufficient to extend the testing to 1083 nm from the 600 nm maximum selection diameter we used.

## 2.2   Using the UHSAS in ORACLES

The aerosol sizing package aboard the NASA P-3B during ORACLES included two DMA systems, the UHSAS, a TSI 3321 Aerodynamic Particle Sizer (APS), and DMT SP2. In addition, a pair of TSI 3563 3-wavelength nephelometers measured aerosol light scattering, which is strongly related to particle size. The second nephelometer was in series with the first and a 1 µm aerodynamic diameter impactor was periodically switched in between them. Sample air for all instruments was drawn through a shrouded inlet that samples aerosol particles with near 100 % efficiency to roughly 3 µm (McNaughton et al., 2007).

Both DMA units used grab samplers (Clarke et al., 1998) to ensure constant size distributions through the 60 second scans. Modified versions of the software developed by Zhou (2001) controlled the systems and inverted the size distributions. One of the units was the modified TSI 3071 LDMA mentioned above. The other was a thermal tandem DMA system (TDMA) that used a nano-DMA (TSI 3085) and a radial DMA (RDMA) (Zhang et al., 1995). It could scan with either DMA after passing sample air through unheated, 150°C, or 300°C thermal denuders or could be configured to select a mobility size with one DMA and scan the resulting particles with the other either directly or after heating to 300°C (Clarke et al., 2004). Sample air was not dried, but excess air from each DMA was desiccated and recycled as sheath air, so particles in the DMA were rapidly dried as they migrated through the sheath air and sizing was effectively at low RH. Particles passing through the DMAs were detected with TSI 3010 CN counters.

The UHSAS operated in slightly different configurations each year. In 2016, sample air was either unheated or passed through a 400°C thermal denuder that eliminated volatile material (Clarke, 1991). The valve system suffered from electrophoretic losses of small particles (due to short segments of insufficiently conductive tubing), so an empirical diameter-dependent correction was implemented by comparison with the long DMA in field data and confirmed with lab measurements with and without the denuder. A linear dropoff of passing efficiency below $D_0 = 21$ µm approximated the data tolerably well: $E = 1 - 3.667(D_0 - D_p)$ where $D_p$ is the particle diameter. This correction, about a factor of 2 at 0.75 µm, has very minor effects on scattering calculations and aerosol mass, but is important for cloud condensation nuclei and total particle number.

The denuder was absent in 2017, but reintroduced in 2018 with the phoretic loss problem fixed. In 2017, sample flow to the UHSAS was diluted 50:50 with desiccated zero air to ensure a low RH, reducing particle growth due to liquid water to a minimum. High RH was only rarely a problem, and only in the marine boundary layer, as the FT had low water vapor content and sample air was heated to cabin temperature, so RH even before desiccation was $< 52\%$ about 95 % of the time.

In 2018 the thermal denuder and desiccated dilution flow were used (though the volatility data are not explored here), and as mentioned above, a system was installed to divert size-selected particles from the TDMA to the UHSAS. The dilution flow system was bypassed during this mode of operation as RH was already low from the TDMA. The largest particles the RDMA could select at altitude were 180 nm; the LDMA could select 500 nm particles, but was about 4 m away from the UHSAS, so it was not practical to use for the in-flight calibration.

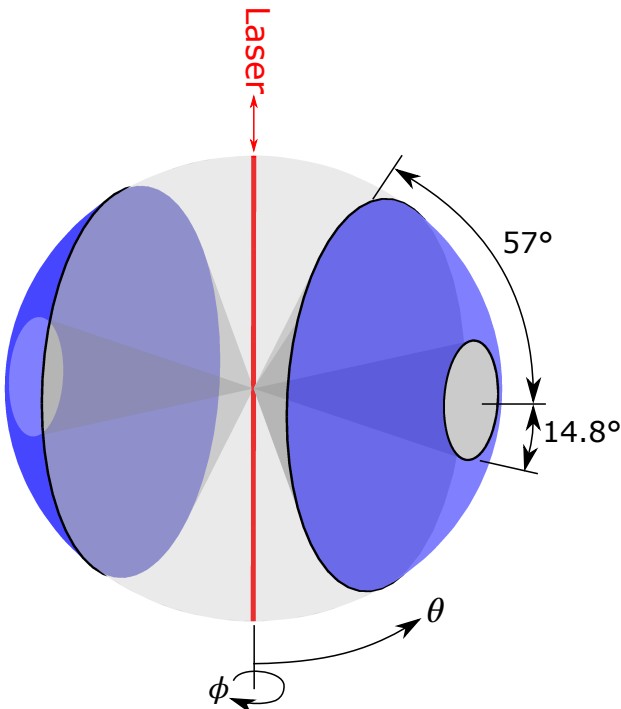

**Figure 1.** Scattering angles detected by the UHSAS. Mangin mirrors on opposite sides of the laser focus scattered light on the detectors from annular regions perpendicular to the beam between 14.8° and 57°(shown as the blue surface). The Mie and MSTM routines calculate scattering integrated over angles $\theta$ (away from the beam) and $\phi$ (relative to the laser polarization) and integrate over the sensing region.

## 2.3 Optical calculations for the UHSAS

In the UHSAS, scattered light is focused on detectors on opposite sides of the scattering region. Each uses a identical set of optics consisting of a pair of Mangin mirrors that accept light scattered in an annular region between 14.8° and 57.0° from perpendicular to the beam (Fig. 1). There must also be a direct 90° scattering path from the particle to the detectors that bypasses the mirrors, but that is a trivial fraction of the collected light and is ignored in these calculations[2]. Laser light is linearly polarized with the electric field parallel to the particle beam and perpendicular to the axis of the collection optics (Droplet Measurement Technologies, 2013).

Mie scattering calculations for most of the particle materials were performed with Matlab code derived from the FORTRAN programs in Wickramasinghe (1973) and checked against other libraries, including Bohren and Huffman (1983) and PyMieScat (Sumlin et al., 2018a). Appendix A has some details about how the calculations were done.

---

[2]The photodiodes are about 5 cm from the beam. If they are 1 cm in diameter (probably an overestimate) then they each subtend a solid angle of 0.031 sr, less than 1.2 % of the 2.65 sr collected by each set of mirrors.

### 2.3.1 Multiple Sphere T-Matrix (MSTM) scattering calculations for rBC particles

Formally, Mie scattering theory applies only to a limited variety of ideal particle shapes: homogeneous spheres with uniform refractive index, homogeneous spheres with spherical coatings, and infinite circular cylinders. This works pretty well even for particles that deviate a bit from sphericity, such as NaCl, which forms cubic crystals but when generated as aerosol is much more spherical than one might expect (Zieger et al., 2017). However, BC is typically so far from spherical that Mie theory is likely to produce inaccurate results. To address this, we used the code described by Mackowski (2014) and available at www.eng.auburn.edu/users/dmckwski/scatcodes. MSTM version 3.0 can be used to calculate scattering due to any arbitrary collection of homogeneous spheres that are either non-contacting, tangent, or entirely contained within each other. Thus, it is well suited to model BC particles as assemblies of spherules, as appears to be largely the case in both ambient aerosol during ORACLES (Miller et al., 2021) and of the fullerene soot used in the lab tests (Moteki et al., 2009). Some details on how MSTM output was used to calculate scattering into the UHSAS optics are in Appendix B.

For this work, we cannot address the infinite variety of shapes that a jumble of spherules can assume, so we made simplifying assumptions that satisfy the data we have. There are two firm constraints when attempting to model fullerene soot: the mobility diameter from the DMA and the mass of rBC as a function of mobility diameter from Gysel et al. (2011). We generate simulated particles using hexagonal close packing to fill a volume with as many 20 nm spherules as possible, discard any spheres whose centers lie outside the mobility diameter, then remove spherules at random until the mass equals that from Gysel et al. (2011). The particle is spun a random amount around each axis to prevent the beam from aligning with what remains of the original lattice. This results in a roughly spherical particle with mobility diameter only slightly smaller than overall diameter (DeCarlo et al., 2004). An example 300 nm particle is shown in Fig. 2. Of course the particles we actually sampled did not look like this–they presumably had more densely packed regions and irregular protrusions, but we believe this simplified model is useful for exploring the optical properties of rBC. More sophisticated modeling with larger deviations from sphericity and proper calculations of aerodynamic resistance will be pursued in later work.

### 2.3.2 Particle heating calculations

Cai et al. (2008) calculated that heating in the UHSAS beam was insufficient to significantly shrink particles of $NH_4NO_3$, which are quite volatile, but absorb IR light very weakly. They confirmed that the UHSAS sized $NH_4NO_3$ accurately, but did not address the far higher absorption of particles containing LAC.

Many processes occur that change the internal energy of a particle as it passes through the beam, including heating though absorption of light, oxidation, and annealing; cooling by conduction to the surrounding air, thermal radiation, and thermionic emission; and mass loss through oxidation and evaporation (Vander Wal et al., 1995; Michelsen, 2003; Michelsen et al., 2007; Bambha and Michelsen, 2015). Below the vaporization point of soot, where evaporation becomes important, the most important processes are light absorption and conductive cooling, so we concentrate on those, ignoring the rest. MSTM calculates absorption, so for BC we use the same model particles and calculations as for scattering while continuing to use Mie calculations for other materials.

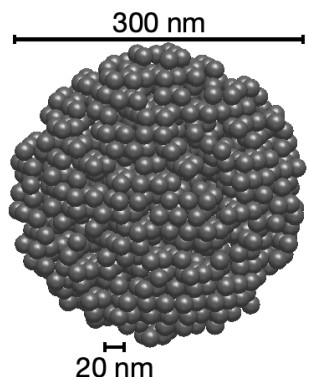

**Figure 2.** A model 300 nm mobility diameter fullerene soot particle made by packing as many 20 nm spherules as possible into a 320 nm spherical region, then removing random spherules until the mass agreed with the formula in Gysel et al. (2011)

Bambha and Michelsen (2015) demonstrated that careful modeling of rBC behavior in an SP2 could duplicate observed scattering and incandescence signals with few arbitrary parameters. A similar model could be applied to the UHSAS, but it lacks the fast data acquisition and incandescence channels needed to verify the model well. Our goal here is more modest; just to establish whether the particles are likely to heat sufficiently to begin evaporating. To do that, we calculate the particle temperature required for absorption from the laser to balance conductive loss to the surrounding air. That can be expressed in the following equation, which is based on Bambha and Michelsen (2015) and derived in Appendix C:

$$T_p = \frac{IQ_{\text{abs}}}{8\kappa_a}\left(D_p + \frac{8fL}{\alpha(\gamma+1)}\right) - T_{\text{air}}. \tag{1}$$

$I$ is the intensity of the laser, $\approx 5.1 \times 10^9 \, \text{Wm}^{-2}$ (Cai et al., 2008). $Q_{\text{abs}}$ is the absorption efficiency of the particle, $\kappa_a$ is the thermal conductivity of air, $f = 2.03$ is the Eucken correction to thermal conductivity, $L$ is the mean free path of the air molecules, $\alpha$ is an accommodation coefficient, and $\gamma = 1.4$ is the ratio of the heat capacities of air at constant pressure and at constant volume.

Particles spend about $\sim 20\,\mu\text{s}$ crossing the beam. The time constant for heating is

$$\tau = D_p^2 \rho_p c_p / 12 k_a \tag{2}$$

where $\rho_p$ and $c_p$ are the density and heat capacity of the particle (Cai et al., 2008). This is approximately $6\,\mu\text{s}$ for $1\,\mu\text{m}$ particles, and shorter for smaller diameters, so particles should essentially attain steady state temperatures (unless the temperature is sufficiently high to evaporate particles).

Of course the real situation is much more complicated than this simple equilibrium calculation. Laborde et al. (2012) have a nice example of this process in an SP2. As absorbing particles enter the beam, they heat up. As they do so, the most volatile materials evaporate first, changing the size, shape, and refractive index of the particle. Some of the material may char, turning

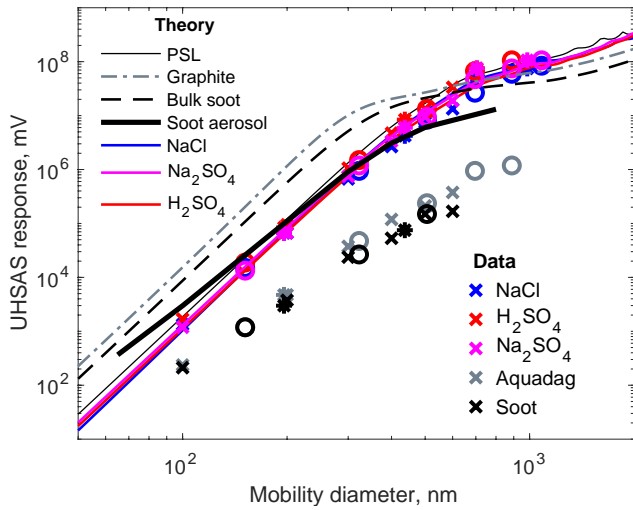

**Figure 3.** Scattering calculations over the collection angles of the UHSAS for particles with various refractive indexes and observed responses. Mie calculations assuming spherical, homogeneous particles are shown except for the soot aerosols, which were assumed to consist of 20 nm spherules as shown in Fig. 2 and discussed in Sec 2.3.1. Symbols represent measured response from particles sized with a long DMA. '×' are singly charged, '○' are doubly charged, and '∗' are triply charged particles. For test particles the y axis is the signal from the peak height detector. For the theoretical curves it is $V$ from Eq. A7 with $C_{\text{opt}}$ determined from the peak height of sub-300 μm PSL calibration spheres. The "soot" data are from the same fullerene soot lot as in Gysel et al. (2011)

into something like rBC (Sedlacek et al., 2018b). As the particle enters more intense regions of the beam, the additional scattering is counteracted by the shrinking particle, making the UHSAS underestimate the particle size. If the particle heats to incandescence, the emitted light may contribute up to 25 % of the light detected (Bambha and Michelsen, 2015), making the particle appear bigger.

## 2.4 Calculated scattering for a variety of materials

Calculated UHSAS response to several types of particles are shown as lines in Fig. 3. Table 1 shows the corresponding refractive indices. The non-absorbing materials have slightly lower $n$ than PSL so they lie just underneath the almost completely hidden PSL trace. NaCl particles are somewhat oversized by the DMA since they are nonspherical; a size-dependent correction from Zieger et al. (2017) for nebulized NaCl was used here. That correction may be counterproductive, as the nonsphericity enhances sidescatter, but the corrections are relatively small. $Na_2SO_4$ also forms nonspherical particles, but we have not found corrections, so no mobility diameter corrections have been made. Another small potential issue is that sulfuric acid is intensely hygroscopic; there is inevitably some water in the particles despite the low humidity, so the refractive index is not exactly known.

In contrast to the non-absorbing salts, spheres with the refractive indices of fullerene soot and graphite have elevated $n$ and $k$, and scatter far more light up to about 600 nm. Coincidentally, the predicted UHSAS response to model soot particles

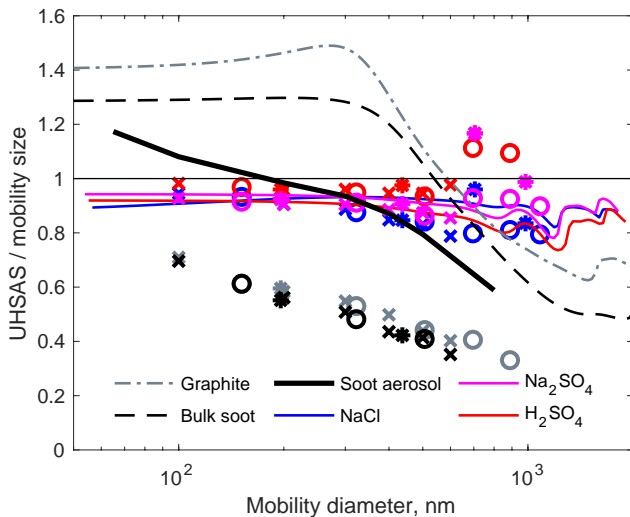

**Figure 4.** The same data as Fig. 3, but the y axis is the ratio of theoretical UHSAS response (lines) or measurements (symbols) to that of PSL spheres.

calculated with MSTM lies almost atop the non-absorptive materials. The void space comes close to canceling the elevated refractive index.

Fig. 4 shows the same scattering calculations recast to show how much bigger or smaller particles of different materials would appear when using a PSL calibration. This magnifies the difference between curves so one can see the complex behavior for particle diameters near the wavelength of the UHSAS laser.

## 3   Results and Discussion

### 3.1   Lab Results

The symbols in Figs. 3 and 4 show the results of the lab tests. In general, the non-absorbing materials follow the expected curves fairly closely, though at diameters above the Rayleigh scattering regime, the data are a bit noisy, particularly between 0.7 and 0.9 µm. This is presumably due to a combination of non-spherical salt particles, Mie wiggles, and for the $Na_2SO_4$, a refractive index at the wrong wavelength. Overall, for this collection of non-absorbing particles, knowing the refractive index reduces sizing errors for particles $< 300\,\mathrm{nm}$ from about 10 % to $\sim 5$ %. For larger particles, the errors can exceed 20 %. It is curious that NaCl is systematically undersized despite the attempted correction, while $H_2SO_4$ is not. The most likely explanation for that is non-sphericity, since the $H_2SO_4$ particles were almost certainly spherical liquid. It may be that the nebulizer we used did not perform like the one used by Zieger et al. (2017) or that the drying rates were different. The particle-to-particle variability of the NaCl particles was higher, yielding less sharp peaks, which might be another effect of non-spherical particles.

**Table 1.** Refractive indices near 1054 nm

| Species | Refractive index |
|---|---|
| PSL | 1.572 |
| $H_2SO_4$[a] | $1.426 - 1.36 \times 10^{-6}i$ |
| NaCl[b] | 1.5314 |
| $Na_2SO_4$[c] | 1.468 |
| graphite[d] | $3.2397 - 2.0233i$ |
| ambient soot[e] | $2.26 - 1.26i$ |
| fullerene soot[e] | $2 - 1i$ |
| smoldering peat[f] | $1.56 - 0.002i$ |
| SAFARI 2000[g] | $1.54 - 0.0094i$ |
| ACE-Asia tarballs[h] | $1.77 - 0.19i$ |
| Lab tarballs[i] | $1.70 - 0.062i$ |

a) 95 % $H_2SO_4$ in $H_2O$ (Palmer and Williams, 1975)

b) Li (1976) via https://refractiveindex.info

c) Kroschwitz (2004), though crystal form, orientation, and coordinated $H_2O$ could give $n$ from 1.394 to 1.483 (Lide, 2004). These are all at $\lambda = 589\,\text{nm}$, as IR refractive index could not be found.

d) Djurišić and Li 1999

e) Moteki et al. 2010

f) Sumlin et al. 2018b at 1047 nm

g) from Haywood et al. (2003) at 550 nm but $k$ reduced from $0.018i$ assuming $1/\lambda$ dependence of absorption.

h) Asian outflow (Alexander et al., 2008)

i) Lab generated from spruce and locust (Hoffer et al., 2017), extrapolated from 950 to 1054 nm and averaged.

The two varieties of black carbon scatter far less light than the calculations suggest, from an order of magnitude for small particles to a factor of 100 for the largest particles. This results in undersizing from 30 % at 100 nm to 67 % at 900 nm. Dramatic undersizing of absorbing particles by OPCs has been noted before (Whitby and Vomela, 1967), but those were for particles much larger than the wavelength and the undersizing was predicted from Mie theory. (In their case, the undersizing was not as extreme as predicted, which they attributed to non-sphericity.) It is striking that despite very different morphologies (flakes vs. spheres) the two types of rBC behaved very similarly.

One possible reason for undersizing is that particles shrink while heated by the UHSAS laser. That would not be entirely surprising, since the laser intensity is close to that of the SP2, which heats rBC to incandescence.

The biggest uncertainty in Eq. 1 is the accommodation coefficient $\alpha$, which represents the odds that an air molecule will equilibrate with the temperature of the particle during a collision. The effect of $\alpha$ is illustrated in Fig. 5. Fully annealed graphite

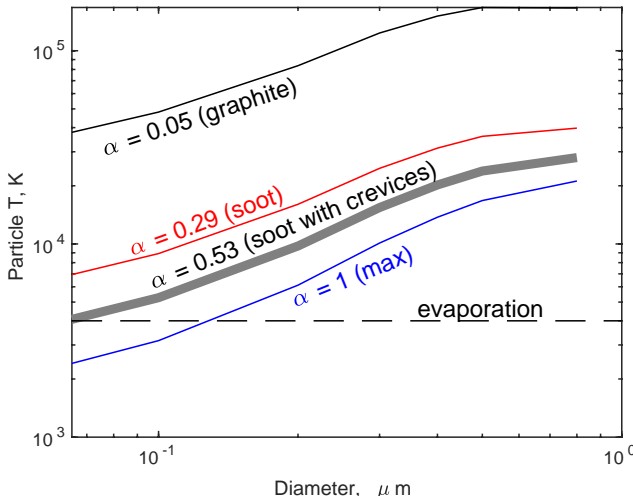

**Figure 5.** Effect of the accommodation coefficient $\alpha$ on calculated heating of model soot aerosol in the UHSAS. Once the particle reaches the evaporation temperature of soot, roughly 4000 K, it will cease heating up and will evaporate until small enough that heating balances conductive cooling.

surfaces are very smooth and have

$$
\begin{aligned}
\alpha_{\text{graphite}} = &\left[0.28 - 3.23 \times 10^{-5}T_p + 0.8e^{-1.53\times 10^{-3}T_p}\right] \\
&\times (0.175 + 5 \times 10^{-4}T_a)
\end{aligned}
\tag{3}
$$

where $T_p$ and $T_a$ are particle and air temperatures, respectively (Michelsen, 2008). At room temperature, before the particle is heated, $\alpha_{\text{graphite}}$ is 0.25, but drops to 0.05 at $T_p = 4000$ K. This inability to transfer energy to air molecules yields exceedingly high calculated temperatures. Even the smallest particles modeled would heat far beyond the $\sim 4000$ K vaporization temperature of graphite and would evaporate almost completely. But soot surfaces are unlikely to be fully annealed and surface roughness are likely to be increased by crystal defects and adsorbed hydrogen, so $\alpha$ is likely to be higher. Bambha and Michelsen (2015) estimated that this increases $\alpha$ by 0.24, yielding $\alpha = 0.29$ at $T_p = 4000$ K. This drops $\Delta T$ by a factor of about 5, but still yields temperatures high enough that that all soot particles should heat to boiling and shrink significantly through evaporation.

Model particles like those shown in Fig. 2 are sufficiently irregular that some incoming air molecules will bounce multiple times as they enter crevices, giving a high chance of equilibrating to the surface temperature. If one assumes that happens to a third of the air molecules incident on the particle, that would increase $\alpha$ to about 0.53, which we consider a reasonable estimate. That would mean particles above about 70 nm would heat sufficiently to begin losing mass.

Even if $\alpha = 1$, minimizing the temperature rise, particles as small as 120 nm would heat to their vaporization point. Larger particles, given enough time in the beam, would evaporate down to that diameter. Altogether, it appears very likely that the fullerene soot and Aquadag are undersized by the UHSAS because the particles evaporate and shrink as they enter the beam.

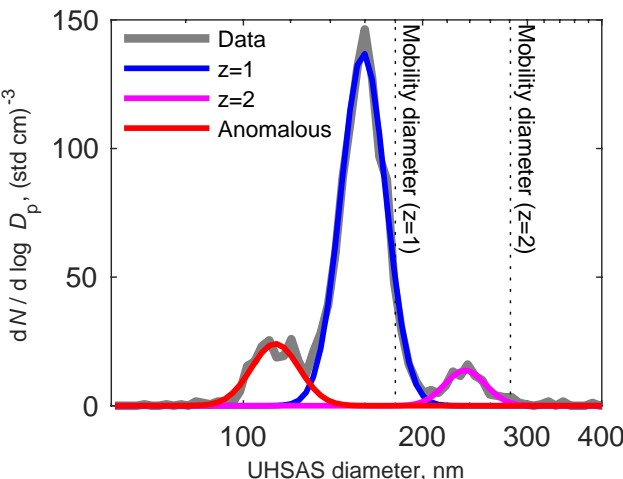

**Figure 6.** UHSAS size distribution from a DMA selecting 180 nm particles at 1.44 °S, 5 °E at 1919 m altitude on 2018/10/23 around 14:00 UTC. Colored curves are lognormal fits to the gray data line.

The actual processes involved are much more complex than the simple equilibrium calculations used here. Crucially, to exhibit reduced scattering, particles must have heated up and partially evaporated before hitting the strongest part of the beam. The peak scattering reflects the interplay between a strengthening beam that increases scattering and energy delivered through absorption, and a shrinking particle that absorbs and scatters less.

### 3.2 Aerosol sampling during ORACLES

In the 2018 ORACLES deployment, we performed the DMA→UHSAS tests during 15 periods, using particles with (singly charged) mobility diameters between 70 and 180 nm. We usually chose 4 diameters and dwelled on each for 60 seconds, then the grab sampler would refill and another cycle would start.

A typical response looks like Figure 6. The biggest peak is plainly the particles selected by the DMA as intended, with a single charge. The width of the fit reflects the transfer function of the DMA (a triangle with curves caused by diffusion), random variation in the UHSAS response, and variations in the scattering properties of the particles.

The peak at larger diameter is due to doubly charged particles. The RDMA could scan up to about 200 nm, depending on altitude; the largest diameter well-defined peak we could get for singly-charged particles was at 180 nm, and corresponding doubly-charged particles were about 280 nm. (The Cunningham slip correction factor was included in the calculation, so mobility is not just the square root of diameter.) Thus we could calibrate only a small fraction of the UHSAS size range, but that included most of the particles in the accumulation mode.

To the left of the singly- and doubly-charged peaks, another mode, containing just a few particles, was present in all of the tests except when the DMA was selecting 80 nm or smaller particles. (That mode may have been present but at diameters below the detection limit of the UHSAS.) This mode appears to be undersized in much the same way as the Aquadag and

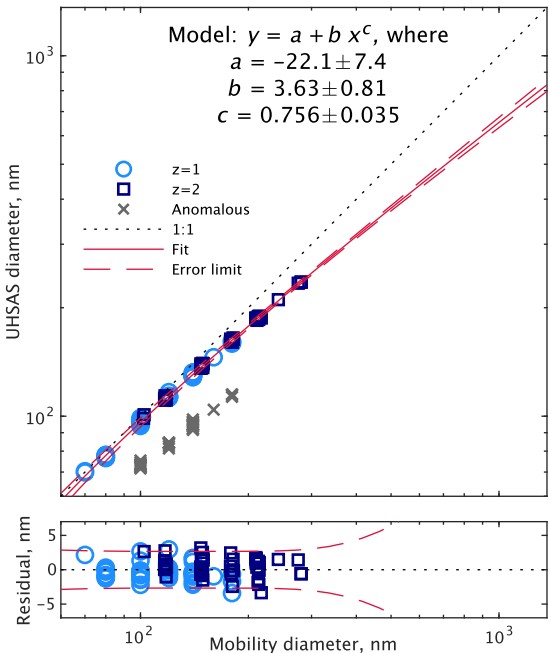

**Figure 7.** UHSAS reported diameters while sampling size-selected particles from the DMA during flight. Only the singly and doubly charged particles were used in the curve fit; the anomalous particles were excluded. (error bounds are 95 % confidence limits within the fitting range). The bottom plot shows the residuals.

fullerene particles in the lab calibrations. Thus, we think it most likely that these anomalously undersized particles are the ones containing rBC. This is discussed later, in section 3.4.

It is obvious in Figure 6 that the diameters reported by the UHSAS for both singly and doubly charged particles are considerably smaller than the mobility diameter. This pattern was consistent through all sizes of particles for each test we did. The top plot in Figure 7 summarizes the results for all of the DMA→UHSAS tests in 2018. The bottom plot shows variability between tests was pretty small, varying no more than $3\,\mathrm{nm}$ from the average. This presumably reflects relatively constant composition through the duration of the 2018 deployment, though we did tend to perform the tests in extended level legs with substantial BB aerosol concentrations, thus perhaps biasing our samples toward constant composition.

We tested a variety of fitting functions with the $z = 1$ and $z = 2$ peaks (omitting the anomalous particles). The simplest function that lacked a strong pattern in the residuals was

$$D_{\mathrm{opt}} = a + bD_{\mathrm{mob}}^c \tag{4}$$

where $D_{\mathrm{opt}}$ is the optical (UHSAS) diameter, $D_{\mathrm{mob}}$ is the mobility diameter from the DMA and $a$, $b$, and $c$ are the fitting parameters. Within the RDMA size range, 95 % confidence limits are within $3\,\mathrm{nm}$, but of course the correction is highly

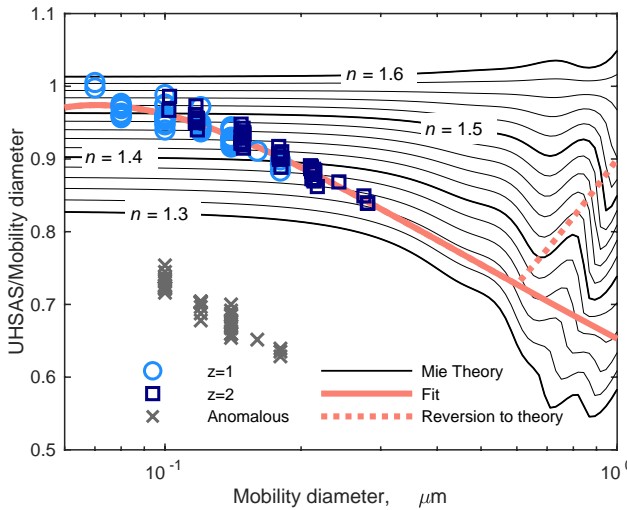

**Figure 8.** The ratio of apparent to actual diameter measured in the DMA→UHSAS tests superimposed on a contour plot of Mie calculations of refractive index and apparent particle diameter. The labels show real refractive indices; absorption is assumed to be 0. The fit line is from Fig. 7, and the dashed line is an arbitrary modification meant to reflect a transition from BB particles in the accumulation mode to seasalt-like particles at 1 μm. Symbols represent the same data as in Fig. 7.

speculative beyond that, from $300\,\mathrm{nm}$ to $1\,\mathrm{\mu m}$. It is likely to be quite inaccurate for particles beyond the accumulation mode,
which are usually chemically distinct from smaller particles, typically dust in the FT and sea salt in the MBL.

The undersizing of the $z = 1$ and $z = 2$ particles in Fig. 7 cannot be explained by the refractive index of the BB plume. For spherical, homogeneous, and non-absorbing particles it is straightforward to determine the refractive index required for particles of a known diameter to produce a given scattering signal. The results are shown in Figure 8. The smallest particles produce scattering consistent with a refractive index of around 1.52, which is lower than usually reported, but not unreasonable.
But the apparent refractive index of larger particles drops rapidly to 1.35, far below any plausible aerosol material. Extrapolating to 0.5 μm gives a refractive index of 1.33, equal to that of water.

There are two obvious explanations for the undersizing. It is possible that the particles are sufficiently aspherical that their behavior is even more pronounced than in the laboratory NaCl tests. That seems unlikely for non-rBC particles since they are a product of vapor deposition to nucleation mode particles, but cannot be excluded. A more likely cause of the undersizing is
evaporation of volatile material from particles as they absorb heat from the UHSAS laser. That is consistent with the exaggerated undersizing for larger particles, which are likely to get hotter, as seen in Eq. 1 and Fig. 5. But what is absorbing IR light from the laser, and how warm do the particles need to get to shrink by 15%?

The thermal denuders on the radial DMA in the TDMA showed that heating to 150°C had little effect, but heating to 300°C reduced particle diameters considerably. The actual mass loss was not practical to quantify, since the radial DMA could not
exceed 200 nm. However, in the tandem DMA tests, when particles were selected with the radial DMA, heated to 300°C, then scanned with the nano DMA, particles typically lost 30% of their mobility diameter, though the range was pretty large (Fig. 9).

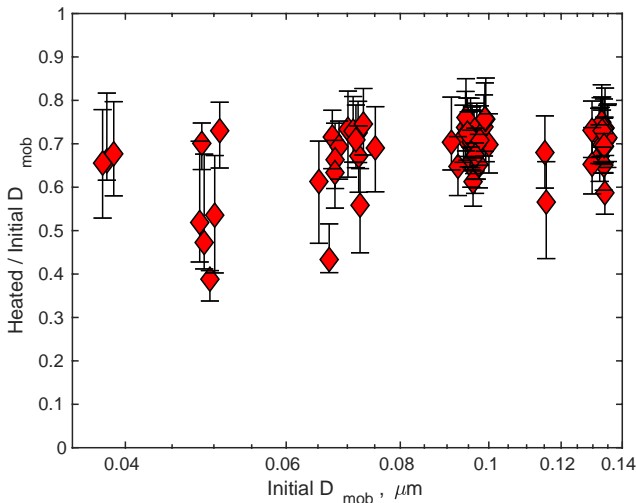

**Figure 9.** Particle volatility at 300°C from the thermal tandem DMA. Symbols are at the median diameter ratio before and after heating. Error bars are the 25th to 75th percentiles, encompassing half of the particles. Data are from all 2018 TDMA measurements in plume (above the marine boundary layer and with nephelometer scattering $> 20\,\mathrm{Mm}^{-1}$).

Since the undersizing of the UHSAS was $\leq 15\%$ (Fig. 8), it is clear that heating to 300°C would be quite sufficient to explain the results. It is curious that volatility depended very little on particle diameter, suggesting that while individual particles at a given diameter were externally mixed, with a range of volatility and hence chemical composition, particle composition differed little across the size ranges reached by the TDMA.

The only significant absorbing materials commonly present in aerosol particles are rBC, dust, and BrC. Since we think the rBC-containing particles appear as the anomalously small mode (see Sec. 3.4), and dust was only occasionally present and is typically on larger particles, BrC is the obvious remaining possibility. We had assumed that that brown carbon would be unlikely to heat this way, as BrC absorption in the IR is weak and is typically considered trivial at long wavelengths (e.g., Yang et al., 2009; Bahadur et al., 2012; Saleh et al., 2013).

Heating calculated from Eq. 1 for a few reported refractive indices of BrC are shown in Fig. 10. The soot aerosol from Fig. 5 is shown for comparison, though the void volume of natural rBC is likely to be different than of the model particles. It is not clear what accommodation coefficient is most appropriate. Winkler et al. (2004) found that $\alpha$ for cloud droplets was indistinguishable from 1, but that may not be the case for organic-rich BB particles at low humidity, so the ranges of calculated heating from $\alpha = 0.5$ to $1.0$ are shown. $200\,\mathrm{nm}$ accumulation mode particles with the reported refractive index of African biomass burning (SAFARI 2000) reach around 300°C, sufficient to evaporate sulfates (Clarke, 1991) and a large fraction of organic material (Ellis and Novakov, 1982; Maruf Hossain et al., 2012) and consistent with the TDMA data. But note that few individual particles are likely to have that refractive index–the real aerosol population is likely to have some particles that contain LAC and thus have much higher absorption while particles lacking LAC absorb little.

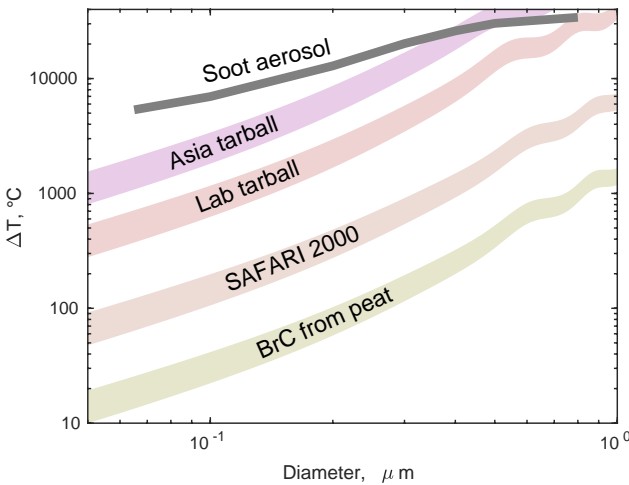

**Figure 10.** Estimates of particle heating from Eq 1 for several particle composition types. MSTM was used to find $Q_{abs}$ for the soot aerosol, while Mie theory was used for the other materials. See Table 1 for the refractive indices used. Shaded regions indicate uncertainty bounds for the accommodation coefficient $\alpha$ ranging from 1 (bottom) to 0.5 (top). The mean free path $L$ was calculated at 700 hPa, near the median pressure during the DMA→UHSAS tests. The soot aerosol line uses $\alpha = 0.53$, as in Fig. 5, but the lower pressure yields temperatures about 1000 K higher.

IR absorption by BrC has been measured in a few studies. While Li et al. (2020) found negligible absorption for wood tar aerosol above 550 nm, Sumlin et al. (2018b) did find a small amount of absorption, $k = 0.002 \pm 0.005$, for 1047 nm light in smoke from smoldering peat fires in a laboratory. That absorption is only sufficient to heat 200 nm particles by about 60 to 100°C (Fig. 10), which would not significantly shrink the particles. It is not clear what relationship the BrC in either circumstance has to the ORACLES aerosol. During ORACLES and the downwind experiment CLARIFY held in 2017, the

relatively low absorption Ångstrom exponent of 1–1.5 between 470 nm and 660 nm (Pistone et al., 2019; Denjean et al., 2020) indicates that BrC contributions to overall absorption at visible wavelengths were small, but does not exclude some IR absorption.

Tarballs are another possibility. They appear to be ubiquitous in biomass burning plumes (Pósfai et al., 2004), including those in southern Africa (Pósfai et al., 2003). Their incidence increases as plumes age over a few hours, but their size does not,

implying that they are primary particles undergoing photochemical aging rather than secondary material condensing (Sedlacek et al., 2018a; Adachi et al., 2019). There is essentially no information available about how they age over the 2 day to 2 week period experienced by the ORACLES plume, but there were some tarballs noted in filter samples taken aboard the plane in 2018 (Michal Segal-Rozenhaimer, personal communication, 2020). There are only a couple of reported complex refractive indices for tarballs at IR wavelengths; they are shown in Table 1. In addition, while not offering a refractive index for diesel-derived

tar, Corbin and Gysel-Beer (2019) found a mass absorption coefficient 1/23 as high as rBC in 950 nm light, suggesting that tarballs absorb more effectively than BrC. Fig. 10 shows that that tarballs are likely to heat between a few hundred and several thousand degrees, sufficient to evaporate most volatile species in the aerosol and perhaps the tarballs themselves.

There are conflicting reports about the behavior of tarballs in SP2s, presumably because of chemical differences or laser strength: Adler et al. (2019) found that tarball-like LAC from lab and forest fires did not incandesce, indicating that they either absorb little at SP2 (and UHSAS) wavelengths or they evaporate before becoming detectable by incandescence ($\sim 3000$ K). In contrast, Sedlacek et al. (2018b) found that pyrolysis-generated tarballs charred and incandesced, but that laser power had a strong effect on charring of other organics (and might for tarballs). Corbin et al. (2019) and Corbin and Gysel-Beer (2019) found that tarballs from detuned diesel engines evaporated but only a small fraction incandesced, and concluded that vaporization temperatures were between 1000 and 3000 K. It is clear that in the UHSAS, tarballs will heat, but the extent of volatilization is uncertain.

One problem with crediting the UHSAS undersizing to tarballs is that there are only two obvious modes in the DMA→UHSAS tests: moderate undersizing and severe undersizing. But if tarballs are numerous but not dominant, there must be three different types of particles with widely varying absorption properties: tarballs, rBC-containing particles and others. To see only 2 modes, one of the following must be true:

**Tarballs behave like other BrC** with similar absorption in the IR, heating, and particle shrinkage. This isn't consistent with reported refractive indices for tarballs or BrC, but those are not well known.

**Tarballs behave like rBC** heating up in the beam and shrinking. Heating to incandescence is not necessary, as Fig. 9 shows that 300°C is sufficient. But that leaves BrC to heat the rest of the particles.

**Tarballs are rare and BrC absorbs IR** That would be inconsistent with (poorly known) BrC refractive indices but consistent with Pósfai et al. (2003), who saw a high fraction of tarballs in moderately aged plumes, but very few in the regional haze around southern Africa. We do not yet have quantitative information about the actual prevalence of tarballs during ORACLES.

**Tarballs dominate** and are responsible for the moderate undersizing mode. This is contrary to Pósfai et al. (2003), but they do speculate that after a few days of aging tarballs may change appearance or collect sufficient secondary material to disguise them. If the material retains the ability to absorb IR, even a small fraction of a tarball remnant would suffice to heat particles enough to shrink in the UHSAS. This may be the most likely explanation, though difficult to confirm.

There are some other possible explanations of the modest undersizing of the majority of particles that are interesting and are likely to occur on occasion but do not appear likely to explain the presence of the two modes. There could be rBC too small for the SP2 to detect, but sufficient to heat particles to a few hundred degrees. Iron has been detected in some of the ORACLES particles, and some crystal forms absorb in the IR. In particular, magnetite ($Fe_3O_4$) has a refractive index of $2.1132 - 0.35780i$ at $1054$ nm (Querry, 1985), so even a small fraction of magnetite on a particle would heat considerably. In contrast, hematite ($Fe_2O_3$), a more common material, has $m = 2.7574 - 0.0110i$ and would not heat significantly. Another possibility is that as particles containing both rBC and non-absorbing organic material heat, mechanical stresses force the rBC to separate from the rest of the particle. The non-absorbing remainder would be smaller, but would not shrink any further in the beam (Moteki and Kondo, 2007). This is particularly easy to imagine if the rBC is in a clump attached to the side of a host particle.

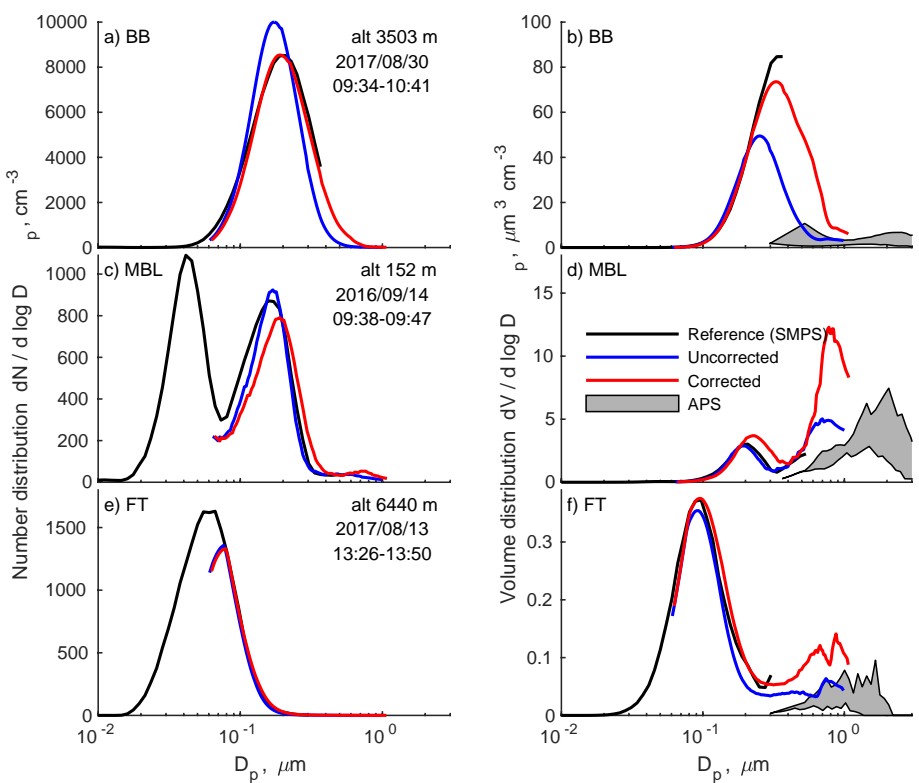

**Figure 11.** Examples of the proposed correction on samples from the polluted FT (a and b), a clean MBL (c and d), and above the plume in the clean FT (e and f). Plots on the left are number distributions and those on the right are volume distributions, which emphasize the larger particles. Since we don't know the density, the APS data are shown as a shaded region between assumed densities of biomass burning aerosol $(1.2\,\mathrm{g\,cm^{-3}})$, and dust $(3\,\mathrm{g\,cm^{-3}})$ in b) and f), or seasalt $(2.2\,\mathrm{g\,cm^{-3}})$in d).

### 3.3   A simple correction scheme

Since the DMA→UHSAS data extend only to 0.28 μm, any extension past that is speculation. When plotting volume distributions, the accumulation mode rarely extended past 0.6 μm. Above that, particles are more likely to be seasalt (in the MBL), dust (in the FT), or sulfate (in a non-dusty FT). In Fig. 8, we show a correction that extends the fit line to 0.6 μm then deviates

430    along a straight line to the expected diameter ratio for particles with $n = 1.54$, similar to $NaCl$, a little above $(NH4)_2SO_4$, and below silicate minerals. This is just a crude way to suppress really exaggerated oversizing for the non-BB particles larger than the accumulation mode, and should not be considered realistic.

Figure 11 shows the effect of applying that correction to some selected size distributions. Uncorrected and corrected UHSAS distributions are compared with LDMA distributions in the pollution layer, the MBL, and clean FT. Data from 2016 and 2017

435    were chosen to demonstrate that even though the correction was based on data from 2018, when BB concentrations were lowest, it is appropriate to use the correction for earlier data in the polluted FT. The top left plot, from a period within the BB

plume, shows that the correction makes the UHSAS agree quite well with the LDMA. The effect on the volume distribution, top right, is dramatic, as the uncorrected data are far below the LDMA distribution. The correction is plainly inappropriate in a clean MBL sample, where the correction overestimates peak particle diameter and accumulation mode volume and is far too high near 1 μm compared with the APS. (Without the transition to NaCl refractive index, it is far worse.) In the clean troposphere, the number peak is at sufficiently small diameters that the correction makes very little difference, so it doesn't matter much whether the correction is used.

When applied to accumulation mode particles in the FT plume, the overall effect of the correction is to increase the median diameter by 13 to 18 nm. In 2018, the least polluted year, the median diameter rose from 143 to 156 nm, while for 2017, median diameter went from 163 to 181 nm with the correction applied.

The danger of extrapolating the correction to diameters above 600 nm even with the arbitrary reversion to $n = 1.54$ is obvious in the volume distributions shown on the right side of Figure 11. Comparisons between UHSAS and APS are not straightforward and are highly sensitive to assumed density, but it is clear that in each of these cases the corrected UHSAS is considerably higher than the APS between 0.6 and 1 μm.

In contrast, scattering emphasizes the large particles less, so the correction fares considerably better, as can be seen in Figure 12, where scattering calculated from the UHSAS distributions is compared to the TSI nephelometer. The correction increases calculated scattering by a factor of about 2.3 in each year, improving agreement considerably. The nephelometers sense scattering between 7° and 170° (Anderson et al., 1996) so the scattering from the corrected UHSAS size distributions were integrated over that range of angles rather than the full 0° to 180° range. This is a more direct comparison than correcting the nephelometer data for truncation errors. See Appendix A for some details.Total rather than sub-μm data are shown because scattering was almost always dominated by the accumulation mode, so the total and submicron nephelometers usually agreed well and over the course of the project we had more data from the total nephelometer. This means the comparison is worse at low altitude, when coarse seasalt particles can account for considerable scattering. This effect is clear, as the dark blue low altitude points are concentrated at the bottom of the data clouds.

While encouraging, the scattering calculation is subject to some major uncertainties, so may have compensating errors:

- Since this nephelometer did not have an impactor removing sub-μm particles, when dust or seasalt were present they increase scattering but are not sensed by the UHSAS. *UHSAS small underestimate.*

- The nephelometer RH was uncontrolled, while in 2017 and 2018 the UHSAS sample was diluted with desiccated air. At high altitudes heating to cabin temperature ensured low RH in all instruments, but nephelometer RH in the MBL occasionally exceeded 50 %. *UHSAS underestimate at low altitudes*

- The same correction is applied to the anomalously undersized particles, which are likely to be 10–35 % of the accumulation mode, as shown below. *UHSAS underestimate.*[3]

---

[3]One could compensate using SP2 data to establish an rBC size distribution, and assume that those particles were undersized as in Fig. 8.

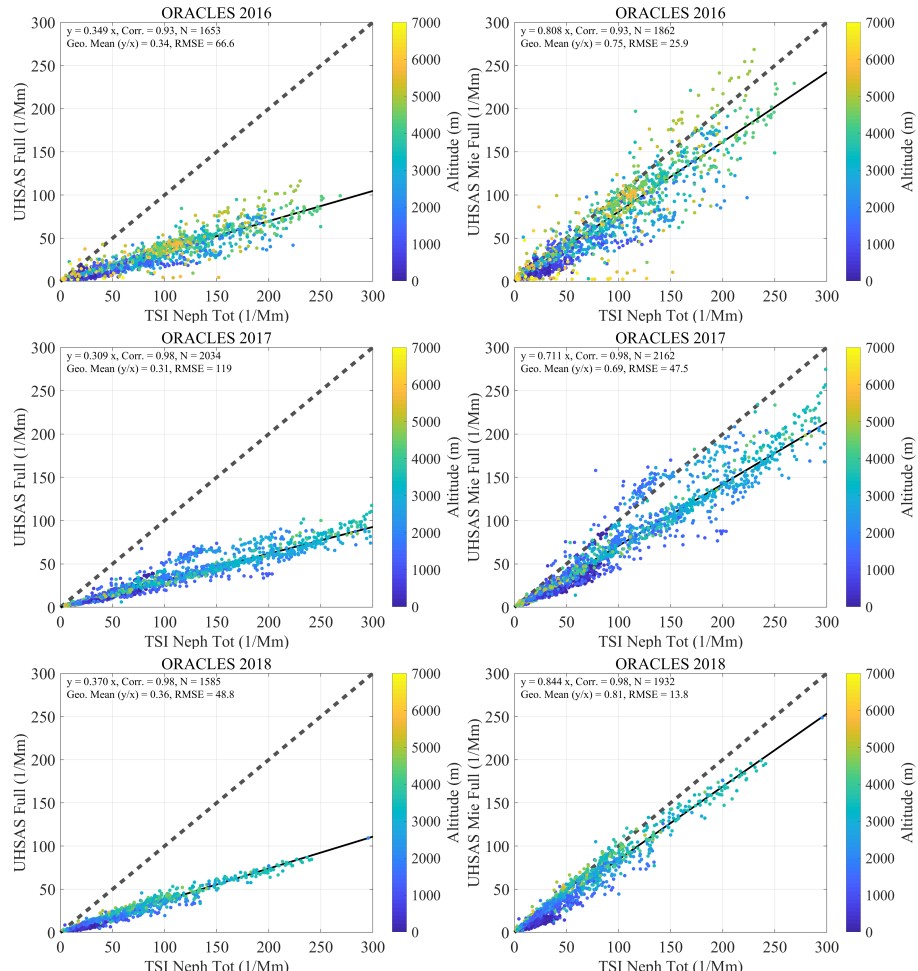

**Figure 12.** Effect of the proposed correction on scattering closure calculations at 550 nm. Plots on the left use uncorrected UHSAS distributions from each year of ORACLES while those on the right use the corrected distributions. Nephelometer data are uncorrected for truncation error; instead the scattering from the UHSAS size distribution is calculated over the 7 to 170 degree sensing range of the nephelometer. The refractive index used in the calculation is 1.588, that of the PSL calibration spheres at visible wavelengths. The 2017 data include a few points above 300 $\mathrm{Mm}^{-1}$ that are not shown.

- We used the PSL refractive index for the Mie calculations, but the correction attempts to produce the correct mobility diameter, rather than an equivalent PSL diameter[4]. The real aerosol presumably has a lower real part and a higher imaginary component. *UHSAS overestimate?*

470

---

[4]For OPCs with lasers near nephelometer wavelengths, closure with nephelometers typically works well if one does no diameter corrections and uses PSL refractive indices, figuring that particles are scattering as if they were PSL particles of that diameter.

## 3.4 The anomalous particles

The particles sized anomalously small in Figs. 6 and 7 are a bit of a mystery. The obvious explanation is that they are rBC particles and behave the same way as the lab tests of fullerene soot and Aquadag (Fig. 3). But the SP2 data indicate that rBC was typically coated with a thick layer of volatile material. In one sense, that makes extensive shrinkage easier to understand–temperatures of 300°C are sufficient (Fig. 9), rather than the 4000°C required to evaporate graphite. But in that case, these shrunken cores ought to be dramatically undersized, as the Aquadag and fullerene soot were. Instead, they look about the same. That may be due to the energy required to evaporate the coating–it may take long enough that the particle has not yet fully shrunk by the time the particle is in the center of the beam. See Laborde et al. (2012) for a nice illustration of that process in the SP2.

It is possible that at least some of these anomalously small particles contain no BC at all, but absorb enough IR that organic materials char and become rBC which then absorbs more IR, heating the particle enough to shrink it a lot. This seems particularly likely for tarballs, as mentioned above. Such charring behavior has been identified in the lab by Sedlacek et al. (2018b), who found that nigrosin particles appeared to an SP2 as rBC with approximately 40 % of the mass of the original particles. That works out to about 70 % of the original diameter, roughly consistent with our anomalous particles (Fig. 8). There were also indications of this in ORACLES when we made a short-lived attempt in 2017 to explore whether volatile coatings on particles enhanced absorption. We planned to test this by passing particles through a 400°C denuder on the way to one of the two PSAPS (3-wavelength Particle Soot Absorption Photometers) on board, anticipating that evaporating some of the coating would reduce absorption. However, the heat treatment increased absorption rather than decreasing it, indicating that charring was taking place.

It would be useful to send size-selected particles to the SP2 and UHSAS simultaneously in future experiments; that would allow us to directly measure the fraction of particles at each size that contain rBC and how much rBC mass was contained in those particles.

The fraction of these anomalous particles is a strong function of particle size (Fig. 13). Below 100 nm, they are either absent or too small to see, but at 180 nm they are over 20 % of the $z = 1$ peak. There were undoubtedly undersized doubly charged particles, but they are hidden by the singly-charged particle peaks. This may mean the number of singly-charged particles is an overestimate and hence the fraction of low scatterers is larger than evident in Fig. 13.

There is also general dependence on altitude, with particles from >3.5 km tending to have fewer anomalous particles than those from lower altitude, though that pattern is a bit noisy. It may be related to other chemical differences we saw with altitude reflecting plume aging (Redemann et al., 2020), but that will require further exploration.

While we cannot prove that the anomalous particles are the ones containing rBC, it is roughly consistent with the overall fraction of rBC-bearing particles shown in Fig. 14. The drastically undersized particles average only 15 to 20 % in the DMA→UHSAS tests, but that may be an underestimate, as mentioned above. In addition, the accumulation mode median particle diameter was roughly 170 nm, averaged over all three years, so the data shown in Fig. 13 only cover half of the size distribution and the relatively high points at the right are the most representative.

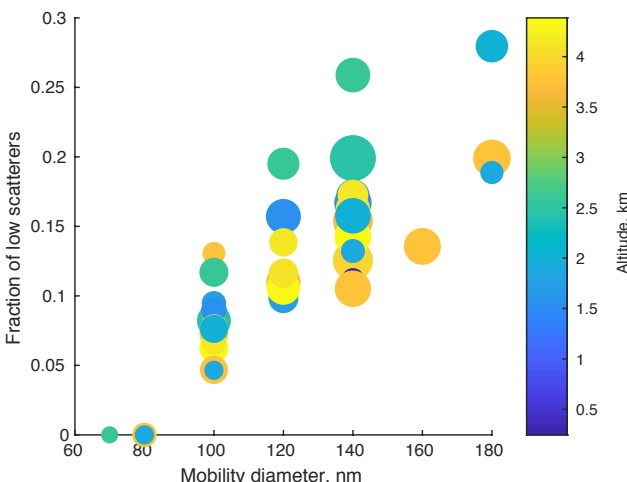

**Figure 13.** Ratio of anomalous to singly charged particles vs. mobility diameter. It is apparent that these anomalously low scattering particles are more common at larger diameters and are somewhat less common high in the plume. Circle sizes are proportional to concentration at that diameter; there is no obvious pattern. The UHSAS lower size detection limit precludes detecting these particles below 70 nm.

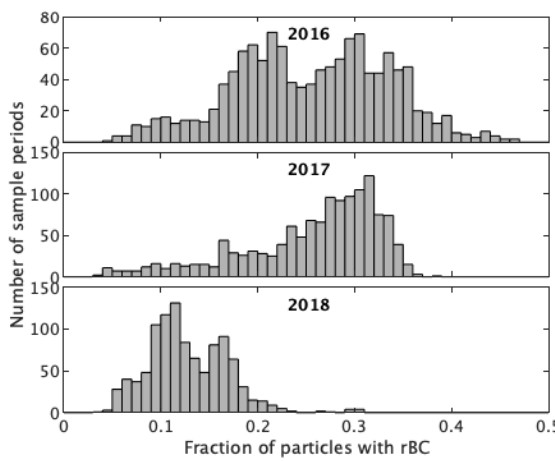

**Figure 14.** The fraction of particles that contained significant rBC in each ORACLES deployment, determined as the ratio of incandescing particles in the SP2 to CN concentration while in the BB plume (altitude $> 1500$ m and rBC concentrations $>50$ cm$^{-3}$). Sample periods were 85 s long, corresponding to an LDMA cycle.

505    Fig 14 also appears to shed some light on the patterns seen in the scattering closure calculations (Fig. 12). Since the anomalous particles are sized roughly 30 % small, scattering calculated from them is less than half of the real value. In 2018, that affected 10–17 % of the particles, and the calculated scattering was $\sim$20 % low. In 2017, by contrast, roughly 30 % of the particles had rBC and calculated scattering was 30 % low. 2016 was in between.

## 4 Conclusions

While for non-absorbing particles, the characteristics of the UHSAS greatly reduce uncertainties due to Mie wiggles, absorbing material complicates the situation, apparently because the very intense IR laser heats the particles. During the ORACLES project over the southeast Atlantic Ocean, the NASA P-3B was often in a fairly uniform biomass burning plume for periods exceeding 30 minutes, allowing time to explore the details of the UHSAS response by selecting single particle sizes with a DMA and passing them to the UHSAS. Two modes of responses appeared. Most particles were moderately undersized, apparently due to heating of brown carbon, though mis-sizing to to non-sphericity cannot be ruled out. A fraction were dramatically ($\sim$30 %) undersized, presumably because they contained more absorbing rBC. It would be interesting in future projects to send size-selected particles to the SP2 as well, so the size-dependent fraction of particles containing rBC could be determined.

Mie calculations using the geometry of the UHSAS showed that the refractive index of the particles was an insufficient explanation of the undersizing. An empirical correction equation $D_{\text{opt}} = a + bD_{\text{mob}}^c$ dramatically improves agreement with DMA distributions between 100 and 500 nm and with scattering closure calculations. This raises median particle diameters between 13 and 17 nm in project average size distributions. The correction is only valid in polluted instances; clean marine boundary layer and free troposphere aerosols behaved more like the calibration spheres. We were unable to directly test the correction between 500 and 1000 nm, though APS data appear to show that the correction fails at the largest diameters, which is no surprise as the composition of those particles are likely to be dust or sea salt, with quite different refractive indices.

Our findings differ a bit from Yokelson et al. (2011), who contend that absorbing particles are essentially invisible to the UHSAS. Instead, the particles are drastically undersized. That leads to complications interpreting size distributions in a rBC-rich plume, when up to 35 % of the particles are likely to suffer this mis-measurement (Fig. 14). Therefore, the relatively slow DMA measurements (one sample every 85 s) must be regarded as the most reliable diameter measurement, with the UHSAS providing rapid response and filling the gap between DMA and APS measurements.

In ORACLES we were fortunate to have numerous redundant sizing instruments available to identify the UHSAS issues. Combustion efficiency for the African fires was quite high, which is typically associated with a high ratio of rBC to organics, so the severely undersized fraction of particles was clearly evident. Such behavior could be important to resolve for other studies using the UHSAS that focus on combustion-derived aerosol and the radiative properties of the size distribution or its relationship to CCN. Moreover, the IR absorption of BrC is little studied, poorly known, and may vary with fuel type, combustion conditions, and aging during transport, so the behavior in the UHSAS may not be the same.

*Code and data availability.* All data from ORACLES are available at the following DOIs:

**2018 P3 data** ORACLES Science Team: Suite of Aerosol, Cloud, and Related Data Acquired Aboard P3 During ORACLES 2018, Version 2, NASA Ames Earth Science Project Office, 2020, Accessible at doi://10.5067/Suborbital/ORACLES/P3/2018_V2

**2017 P3 data** ORACLES Science Team: Suite of Aerosol, Cloud, and Related Data Acquired Aboard P3 During ORACLES 2017, Version 2, NASA Ames Earth Science Project Office, 2020, Accessible at doi://10.5067/Suborbital/ORACLES/P3/2017_V2

**2016 P3 data** ORACLES Science Team: Suite of Aerosol, Cloud, and Related Data Acquired Aboard P3 During ORACLES 2016, Version 2, NASA Ames Earth Science Project Office, 2020, Accessible at doi://10.5067/Suborbital/ORACLES/P3/2016_V2

**2016 ER2 data** ORACLES Science Team: Suite of Aerosol, Cloud, and Related Data Acquired Aboard ER2 During ORACLES 2016, Version 2, NASA Ames Earth Science Project Office, 2020, Accessible at doi://10.5067/Suborbital/ORACLES/ER2/2016_V2

Processing code is available as well. The Mie calculation code in Matlab is available on request.

## Appendix A: Mie scattering calculations

Light scattering by any particle can be represented by 4 amplitude functions, $S_1$, $S_2$, $S_3$, and $S_4$, which are complex functions dependent on the angles of the incident beam and the scattered light. For homogeneous spheres, only $S_1$ and $S_2$ are nonzero (van de Hulst, 1957) and are functions of the Mie parameter $\pi D_p/\lambda$ and the refractive index $m = n - ik$. $D_p$ is particle

diameter, $\lambda$ is wavelength, $n$ is the real part of the refractive index, and $k$ is the complex, absorbing part of the refractive index.

The intensity of light scattered from a single homogeneous spherical particle is proportional to the square of the magnitude of the amplitude functions:

$$I_\perp = \frac{\lambda^2}{4\pi^2 r^2} \left|S_1(\theta)\right|^2 I_0 \tag{A1}$$

for perpendicular polarization and

$$I_\parallel = \frac{\lambda^2}{4\pi^2 r^2} \left|S_2(\theta)\right|^2 I_0 \tag{A2}$$

for parallel polarization, where $\theta$ is the scattering angle, $\lambda$ is the wavelength of light, $r$ is the distance from the particle, and $I_0$ is the intensity of the illuminating beam as power per unit area.

The UHSAS laser is reflected within the cavity, so the beam is going both ways, doubling the intensity:

$$I_\perp = \frac{\lambda^2}{4\pi^2 r^2} I_0 \left[\left|S_1(\theta)\right|^2 + \left|S_1(\pi - \theta)\right|^2\right] \tag{A3}$$

and

$$I_\parallel = \frac{\lambda^2}{4\pi^2 r^2} I_0 \left[\left|S_2(\theta)\right|^2 + \left|S_2(\pi - \theta)\right|^2\right] \tag{A4}$$

Combining Eqs. A3 and A4 using the angle $\phi$ from parallel polarization and substituting $\mathbb{S}_1(\theta)$ and $\mathbb{S}_2(\theta)$ for the quantities in the square brackets gives the total intensity as a function of position:

$$I = \frac{\lambda^2}{4\pi^2 r^2} I_0 \left[\mathbb{S}_1(\theta) \sin^2 \phi + \mathbb{S}_2(\theta) \cos^2 \phi\right] \tag{A5}$$

Power into the detector is the integral of the intensity over the collection area of the optics:

$$P = \frac{\lambda^2}{4\pi^2} I_0 \int_A \frac{1}{r^2} \left[\mathbb{S}_1(\theta) \sin^2 \phi + \mathbb{S}_2(\theta) \cos^2 \phi\right] dA \tag{A6}$$

Realizing that $dA = r^2 \sin\theta\, d\phi\, d\theta$ and multiplying by a constant $C_{\text{opt}}$ that includes the efficiency of the mirrors, the sensitivity of the detector, and the amplification of the output circuit gives the output voltage of the detector as

$$V = C_{\text{opt}}\frac{\lambda^2}{4\pi^2}I_0 \iint_{\theta\,\phi} \left[\mathbb{S}_1(\theta)\sin^2\phi + \mathbb{S}_2(\theta)\cos^2\phi\right]\sin\theta\, d\phi\, d\theta \tag{A7}$$

It is assumed in Eqs. A3 and A4 that the UHSAS is not an active cavity device–beams going each direction are not coherent; they do not interfere with each other and no standing wave pattern is present. If it were, scattering from individual particles would depend on precisely where the particle meets the beam and on average would require summing the scattering amplitudes before calculating intensity (Garvey and Pinnick, 1983). This just means redefining $\mathbb{S}_1$ and $\mathbb{S}_2$:

$$\mathbb{S}_1(\theta) = |S_1(\theta) + S_1(\pi - \theta)|^2 \tag{A8}$$

and

$$\mathbb{S}_2(\theta) = |S_2(\theta) + S_2(\pi - \theta)|^2 \tag{A9}$$

While the DMT PCASP (Passive Cavity Aerosol Spectrometer Probe) has a vibrating mirror to suppress these standing waves (Rosenberg et al., 2012), the UHSAS does not actively avoid coherence. The manufacturer reports that it is "not a single-mode laser and the coherence time is quite small" (personal communication with DMT, 2017). This is unlike the TSI 3340 OPC, which has detection optics very similar to the UHSAS but an active-cavity HeNe laser ($633$nm) where the manual mentions a standing wave mode.

The implementation is straightforward and fairly primitive: given the refractive index $m$, wavelength $\lambda$, and a vector of particle diameters, $S_1$ and $S_2$ are calculated for $\theta = 0$ to $\pi$ radians (default resolution is $0.5°$). Then the quantities in the integral of Eq. A7 are calculated at each $\theta$ and $\phi$, and points within the collection region of the optics are summed.

The code used is quite general; it is designed to calculate scattering into arbitrarily oriented conic regions so essentially any single-wavelength OPC can be represented. There are options to define $\mathbb{S}_1$ and $\mathbb{S}_2$ as in Eqs. A8 and A9 for exploring the impact of active cavities or simply as

$$\mathbb{S}_1(\theta) = |S_1(\theta)|^2 \quad \text{and} \quad \mathbb{S}_2(\theta) = |S_2(\theta)|^2 \tag{A10}$$

for instruments without an intracavity laser such as the Grimm and MetONE OPCs and for forward scattering instruments like the Forward Scattering Spectrometer Probe (FSSP) and Cloud Droplet Probe (CDP).

The same equations can be used to simulate nephelometer response to a size distribution. In that case, Eq. A7 is applied at each particle diameter $D_p$ with $I_0 = 1$, $C_{\text{opt}} = 1$, and $\mathbb{S}_1$ and $\mathbb{S}_1$ defined as in Eq. A10. Since the TSI sensing volume is symmetric around the beam axis, the inner integral of Eq. A7 is always evaluated from 0 to $2\pi$, so

$$\int_{\phi} \left[|S_1(\theta, D_p)|^2 \sin^2\phi + |S_2(\theta, D_p)|^2 \cos^2\phi\right] d\phi$$

$$= \pi\left(|S_1(\theta, D_p)|^2 + |S_2(\theta, D_p)|^2\right) \tag{A11}$$

and integrating over the entire size distribution yields

$$B_{\mathrm{sca}} = \frac{\lambda^2}{4\pi} \int\limits_{D_p} N_p(D_p) \int\limits_{\theta} \Big( |S_1(\theta, D_p)|^2 + |S_2(\theta, D_p)|^2 \Big) \, d\theta \, dD_p \tag{A12}$$

where $B_{\mathrm{sca}}$ is the scattering seen by the nephelometer before any truncation corrections are applied.

## Appendix B: MSTM 3.0 Scattering Calculations

The assumption that aerosol particles are single homogeneous spheres is a decent approximation for many purposes, but is
cannot represent BC particles well. Electron micrographs of soot particles typically show that they are collections of roughly
20 nm spheres jumbled together. Fresh particles often show a quasi-fractal branching pattern, while aged particles collapse into
more compact shapes. Much work has been done to explore how this non-sphericity affects optical properties of the particles,
but the effect on scattering into OPC optics has apparently not been addressed theoretically.

A freely available software package, Multiple Sphere $T$-matrix (MSTM) (Mackowski and Mishchenko, 2011; Mackowski,
2014), is well suited to address soot-like particles. It calculates optical properties of arbitrary collection of spheres. The spheres
can be separate, within other spheres, or tangent to each other; the only limitation is that sphere boundaries cannot intersect.
Since the calculations can be quite complex, MSTM was developed for use on parallel computers, though a single-processor
version is supplied that compiled easily on an Apple MacBook Pro running macOS 10.15 using GNU gfortran (version 9.2.0)
via the Homebrew package manager (version 2.2.17).
While running MSTM is surprisingly easy, it does take some manipulation to determine the scattering into UHSAS optics.
The terminology used here is adopted from Mishchenko et al. (2006), which differs somewhat from the more familiar Bohren
and Huffman (1983) and van de Hulst (1957), but is a little clearer.

MSTM does not output the scattering amplitude functions as used in Appendix A. Instead, it reports elements of the $4 \times 4$
element scattering matrix $\mathbf{F}$, which transforms the Stokes vector $\boldsymbol{I}_i$ of incident light to the Stokes vector $\boldsymbol{I}_s$ of the scattered
beam.

Stokes vectors are $4 \times 1$ column matrices which can be defined in a number of ways, but MSTM uses a common form
where $\boldsymbol{I} = \begin{bmatrix} I & Q & U & V \end{bmatrix}^\mathsf{T}$ with $I$ representing light intensity, $Q$ parallel and perpendicular polarization, $U$ for polarization
at 45°and -45°, and $V$ circular polarization (Mishchenko et al., 2006). $Q$, $U$, and $V$ are constrained by $I^2 \geq Q^2 + U^2 + V^2$,
with the equality holding only when the light is fully polarized.
The coordinate system with respect to the UHSAS that we use here defines the $z$ axis along the incident laser beam, the
$x$ axis along the particle beam, and the $y$ axis through the centers of the detection optics. Since the electric field vector of
the UHSAS laser is parallel to the particle beam, the Stokes vector for the laser is $\boldsymbol{I}_i = [1, -1, 0, 0]^\mathsf{T}$. The scattering matrices
returned by MSTM use a different coordinate system, where the $xz$ plane is defined by the incident and scattered light beams,
so the laser coordinate system has to be rotated by $\phi$. This is done with transform matrix $\mathbf{L}$ (Mishchenko et al., 2006). Thus,
the complete calculation becomes

$$\boldsymbol{I}_s = \frac{1}{r^2} \mathbf{F}(\theta, \phi) \times \mathbf{L}(\phi) \times \boldsymbol{I}_i \tag{B1}$$

or fully expanded,

$$
\begin{bmatrix} I_s \\ Q_s \\ U_s \\ V_s \end{bmatrix} = \frac{1}{r^2} \begin{bmatrix} F_{11} & F_{12} & F_{13} & F_{14} \\ F_{21} & F_{22} & F_{23} & F_{24} \\ F_{31} & F_{32} & F_{33} & F_{34} \\ F_{41} & F_{42} & F_{43} & F_{44} \end{bmatrix}
$$

$$
\times \begin{bmatrix} 1 & 0 & 0 & 0 \\ 0 & \cos 2\phi & \sin 2\phi & 0 \\ 0 & -\sin 2\phi & \cos 2\phi & 0 \\ 0 & 0 & 0 & 1 \end{bmatrix} \times \begin{bmatrix} I_i \\ Q_i \\ U_i \\ V_i \end{bmatrix}
$$

where $r$ is distance from the particle. Since the UHSAS photodiodes do not detect polarization, only the first term, $I_s$, of the scattered light Stokes vector is of interest.

Another complication is that $F_{11}$ values produced by MSTM are scaled to satisfy

$$
\frac{1}{4\pi} \int\limits_{0}^{2\pi} \int\limits_{0}^{\pi} F_{11}(\theta, \phi) \sin \theta \, d\theta \, d\phi = 1 \tag{B2}
$$

so $I_s$ is not an absolute value. MSTM does report total scattering efficiency, $Q_{\mathrm{sca}}$, and a volume-equivalent radius $r_{\mathrm{v}}$, so the total scattering is

$$
I_t = I_0 \pi r_{\mathrm{v}}^2 Q_{\mathrm{sca}} \tag{B3}
$$

and the fraction of scattering scattering into the UHSAS optics can be determined by integrating $I_s$ over the UHSAS optics, dividing by the integral over the entire sphere. Multiplying by $C_{\mathrm{opt}}$ gives the voltage response expected from the UHSAS:

$$
V = C_{\mathrm{opt}} I_0 \pi r_{\mathrm{v}}^2 Q_{\mathrm{sca}} \frac{\iint\limits_{\mathrm{UHSAS}} I_s(\theta, \phi) \sin \theta \, d\phi \, d\theta}{\iint\limits_{\mathrm{sphere}} I_s(\theta, \phi) \sin \theta \, d\phi \, d\theta} \tag{B4}
$$

One last detail is that MSTM assumes a single beam of light, while the UHSAS laser propagates both ways along the $z$ axis. Since MSTM returns the scattering matrix rather than the amplitude functions, one cannot model active cavity devices where the two beams are coherent. That is not the case for the UHSAS, so we just add the scattering from a particle with that of its mirror image (reflected at the $z = 0$ plane). Results shown here were calculated that way even though scattering results were almost indistinguishable from multiplying by 2 for the nearly spherically symmetric particles we used. However, it did reveal the absorption patterns shown in Fig. B2.

## B1   Comparing MSTM and Mie calculations

As mentioned in the main text, the particles we modeled for the MSTM calculations were roughly spherical so the mobility diameters would be known. Therefore, one might expect differences between Mie and MSTM calculations to be fairly small,

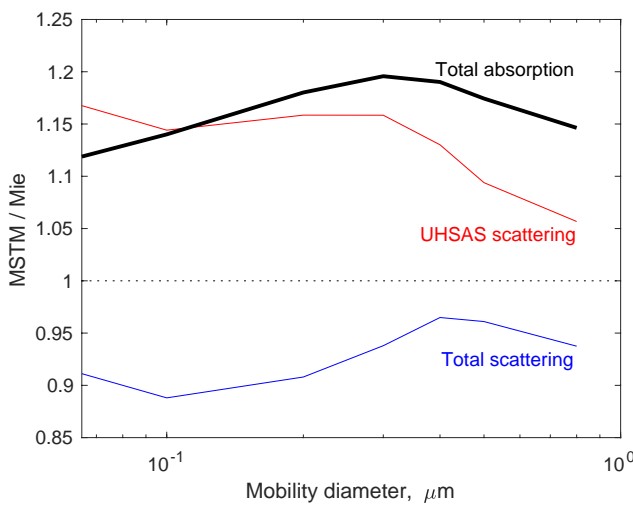

**Figure B1.** Comparison of scattering and absorption calculated from Mie theory with that from MSTM v3.0. Mie calculations assumed spheres with diameter equal to mobility diameter and refractive indices were volume-weighted averages of soot and air (Liu and Daum, 2008). Soot volume was determined from Gysel et al. (2011). MSTM calculations assumed particles were assemblies of 20 nm spheres equaling the masses derived from Gysel et al. (2011) all packed into a spherical volume with diameter equal to mobility diameter plus 20 nm.

and that is seen in Fig. B1. The MSTM calculations show that total scattering is roughly 5 to 10% smaller than Mie calculations would indicate, but that scattering into the UHSAS optics is actually enhanced by 5 to 15%. That yields diameter errors $< 3\%$, since sizing errors go with the 6th root of scattering error.

Somewhat more important is that MSTM calculates greater absorption than Mie theory, by 10 to 20%. That is directly related to the temperatures reached, according to Eq. 1.

Since MSTM calculates the absorption efficiency of each individual spherule, it can reveal heating patterns. Fig. B2 shows spherule absorption efficiencies in 40 nm slabs through model particles of various sizes, and it is clear that for particles from 200 nm to 500 nm there is a focusing effect–spherules near the center of the particles are heated up to 3 times as much as those near the surface. For the 800 nm model particle, the center is shaded and absorption is enhanced where the beams enter the particle. It is striking that the focusing effects clearly involve very small-scale details; neighboring spherules can have very different absorption efficiencies. This is in apparent contrast to the commonly used Rayleigh-Gans approximation often used for soot particles, which assumes that the spherules are affected only by the incident beam Mishchenko et al. (2006) so will all heat the same way.

It is not obvious what effect this uneven heating would have on these model particles, much less real ones, and we have made no attempt to do so. Conduction by interstitial air is likely to be severely hampered by tortuous pathways, so if contact area between spherules is small, limiting heat conduction, thermal radiation may be the primary means of getting heat to the exterior of the particle. If so, small regions within the particle may heat extremely quickly.

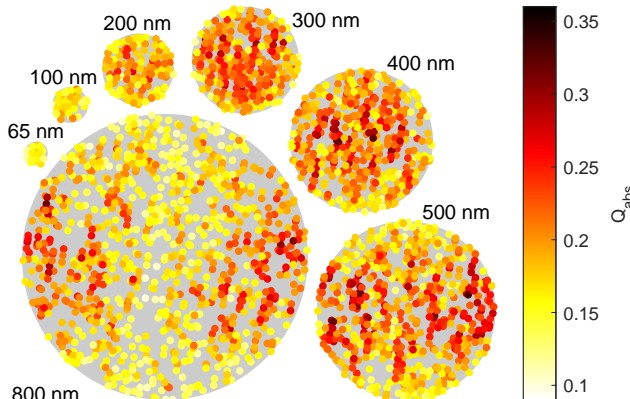

**Figure B2.** Light absorption by individual spherules in the model fullerene soot particles from the MSTM calculations. The laser beams are traveling left and right. Each particle illustration shows a 40 nm slab through the middle of the particle. The yellow to red colors indicate relative light absorption rates, while the gray backgrounds show the mobility diameter. The low effective densities as found by Gysel et al. (2011) are particularly obvious in the large particles.

## Appendix C: Heat transfer from particle to air

For this calculation, we use the equations of Bambha and Michelsen (2015), but do not attempt a detailed time course of particle heating. Instead, the goal is to determine whether particles are likely to reach the roughly 4000 K temperature at which rBC particles vaporize Schwarz et al. (2006) and shrink. Hence we only look at equilibrium temperatures. Air is assumed to be at 670    25°C.

Since particles sizes are on the order of the mean free path $L$ of air molecules ($\sim 65$ nm at room temperature and pressure), conductive cooling occurs in the transition between the free molecular regime ($D_p \gg L$) and continuum conditions ($D_p \ll L$). There are several calculation schemes to address the transition, none of which are completely accepted (Seinfeld and Pandis, 2006). In general, they posit a free molecular layer of thickness near $L$ surrounding the particle and continuum conditions 675    beyond that. Following Bambha and Michelsen (2015), we adopt the method of McCoy and Cha (1974), but use it in a somewhat different fashion, treating the entire particle as a single entity rather than calculating heat transfer from each spherule (which requires determining what area fraction of each spherule is exposed to the air.) Since the scale of the roughness of our model particles is $\sim 20$ nm, much smaller than the mean free path, the particles will effectively act as spheres.

The conductive heat loss rate $\dot{Q}_{\text{air}}$ as a function of the temperature difference $\Delta T$ between particle and surrounding air is

680    $$\dot{Q}_{\text{air}} = \frac{2\pi \kappa_a D_p^2}{D_p + GL} \Delta T \tag{C1}$$

where $\kappa_a$ is the thermal conductivity of air and $G$ is a geometrical term that for a sphere is (McCoy and Cha, 1974)

$$G = \frac{8f}{\alpha(\gamma + 1)} \tag{C2}$$

with an accommodation coefficient $\alpha$, and $\gamma$ is the ratio of heat capacities at constant pressure ($C_p$) and volume ($C_v$). $\gamma = 1.4$ for air at room temperature. $f$ is the Eucken correction to the thermal conductivity, which attempts to take into account the vibrational and rotational characteristics of gas molecules as well as translational speed by splitting heat capacity and corrections into translational and internal (vibration + rotation) modes.

$$f = \frac{C_{v,trans} f_{trans} + C_{v,vibrot} f_{vibrot}}{C_v}. \tag{C3}$$

Here $C_{v,trans} = \frac{3}{2}R$, $C_{v,vibrot} = C_v - \frac{3}{2}R$ (Lyusternik and Mustafaev, 1976), $f_{trans} = \frac{5}{2}$ and $f_{vibrot} = 1.328$ (Istomin et al., 2014).

At equilibrium, heat dissipated from the particle to the air equals the energy absorbed from the laser:

$$\dot{Q}_{air} = I_0 Q_{abs} \frac{\pi}{4} D_p^2 \tag{C4}$$

Combining Eq. C4 with Eq. C1 and Eq. C2 and solving for $\Delta T$ yields Eq. 1

$$\Delta T = \frac{I_0 Q_{abs}}{8\kappa_a} \left( D_p + \frac{8fL}{\alpha(\gamma + 1)} \right) \tag{C5}$$

*Author contributions.* N.S. performed the lab tests and operated the in-flight instrumentation. S.F. helped design the experiment, participated in lab work and flight work, did much of the data processing and extended analysis, submitted the data to the archive, and helped with paper writing. S.H. operated the equipment, helped design the experiment, did many of the calculations, and wrote the paper. A.S. operated the SP2 and processed the data and contributed editorial content. A.D. also operated in-flight instrumentation and provided help in the lab.

*Competing interests.* The authors declare that they have no conflict of interest.

*Acknowledgements.* We are grateful to the entire NASA Wallops aircraft team: pilots, air crew, techs and engineers who worked with the P-3B. In addition NASA's ESPO group provided extremely valuable logistical support. We'd also like to thank the whole ORACLES team for uniting to make such a pleasant and productive project. Funding was from NASA grant NNX15AG01G.

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
