# Peer review of "Undersizing of Aged African Biomass Burning Aerosol by an Ultra High Sensitivity Aerosol Spectrometer"

_Atmospheric Measurement Techniques, 2020_

## Referee Comment (RC1) · Charles Brock (Referee) · 10 Nov 2020

Review of "Undersizing of aged African biomass burning aerosol by an ultra high sensitivity aerosol spectrometer" by S. Howell et al.

This manuscript is a well-written and clear analysis of surprising results from an ultra-high sensitivity aerosol spectrometer (UHSAS) found during the ORACLES airborne project. The size distributions measured by this optical particle counter substantially under-predicted directly measured aerosol scattering and size distributions measured by a scanning mobility particle spectrometer. By introducing size selected particles into the UHSAS, the authors found that 10-30% of the particles measured in biomass

burning plumes downwind of Africa were being undersized. Extensive laboratory investigation led to the conclusion that the UHSAS instrument heats brown carbon or "tarball" particles found in biomass burning plumes, causing partial evaporation and substantial undersizing. The implication is that all UHSAS measurements made in aerosols with a substantial absorbing component must be treated with caution.

Since the UHSAS is widely used in both airborne and ground-based measurements, especially in cases of rapidly changing aerosol conditions where SMPS response is too slow, this manuscript's findings are important. The scientific finding that brown carbon or tarball particles exhibit significant light absorption into the infrared is also of interest to readers.

This manuscript is quite clear, logical, and thorough. The graphs are (mostly) clear, there is a good mix of airborne observations, laboratory experiments, and theoretical analysis, and the data have been made available for public use. This paper is acceptable for publication in AMT with only minor technical edits.

Comments/edits:

1) Line 66. You may want to cite Kupc et al. (2018) here. Kupc et al. describe modification and calibration of a UHSAS for airborne use, and is one of only two (now three) papers discussing the performance of the UHSAS.

2) Line 106. You mention later (line 139) that a "grab" sampler was used by the SMPS. Why do you need a "vast" and apparently homogeneous plume to do the size-resolved analysis? Doesn't the grab sample eliminate the need for homogeneity?

3) Figures 2-5. These figures are well laid out, but they use very similar colors and line types. The colors for the graphite, Aquadag, and NaCl are too similar, as are the sulfate types. I'm not color-vision-impaired, but I know many who are, and these figures would be really tough to read. Can you use more distinctive colors and different line types (dotted, dashed, dash-dotted, etc.) to better distinguish the curves?

4) Line 236. You state that the SP2 has a "much more powerful IR laser". Stephens et al., (2003), who describe the prototype for the SP2, quote a laser intensity of 10ˆ6 W/cmˆ2. This is not very different from the stated (line 204) UHSAS laser intensity of ~5x10ˆ9 W/mˆ2 (= 0.5x10ˆ6 W/cmˆ2). Thus I would not be surprised at all if the UHSAS is able to at least partially incandesce BC particles, leading to a mis-sizing.

5) Line 357. The corrected UHSAS data are not "an order of magnitude too high near 1 $\mu$m" for the free troposphere case (Fig. 8f). It's less than a factor of 2.

6) Line 373. Here you say that it's not clear which refractive index to use to calculate scattering for comparison with the TSI nephelometer. Each bin of the UHSAS correspond to a certain amount of scattering into the detection volume. If you quantify how much scattering each bin represents, which you are effectively doing by calibrating with a monodisperse aerosol, you should just use the same calibrant refractive index to calculate total scattering. In effect, you are just summing up the scattering represented by each bin, getting the total scattering. Of course, this ignores the difference between hemispherically integrated scattering vs. the narrower viewing angles of the UHSAS detection optics, but it is a very good first approximation to just use the refractive index of the calibrant (in this case, PSL). You can investigate the magnitude of the error due to the scattering geometry using Mie calculations.

7) Line 371. Can you use the rBC number fraction from the SP2 to estimate the number of anomalously undersized particles, and boost the number in the main mode by this fraction to compensate?

8) Line 384. The obvious explanation is that the coatings are not volatile at 400 C. This is consistent with Adler et al., 2019, who found coatings on biomass burning particles that did not evaporate (at lower temperature) but that were not incandescent in the SP2.

9) Fig. 9. Have you modified the UHSAS flow system as in Kupc et al.? We found that both of our UHSAS instruments leaked through the seals around the detectors,

downstream of the detection region, reducing the sample flow even though the exhaust flow (which is the nominal flow measurement) was constant. This produced an altitude-dependent flow bias (Brock et al. https://doi.org/10.5194/acp-11-2423-2011), although it works in the direction opposite the trend seen here. Droplet Measurement Technologies has repaired the leak in our UHSAS, and they had a jig and setup to do this, implying that this is a common problem that they have had to fix in the past.

10) Fig. 11. Change y-axis label to "Fraction of Particles".

11) The Appendix is very clear and helpful.

12) Please check over the references for consistency with Copernicus formatting guidelines. For example, Clarke and Ellis et al. have capitalized titles, journal names are not consistent, etc. This is a consequence of EndNote-type software, which ALWAYS needs manual checking and correction.

Adler et al.: Evidence in biomass burning smoke for a light-absorbing aerosol with properties intermediate between brown and black carbon, Aerosol Sci. Technol., 976-989, https://doi.org/10.1080/02786826.2019.1617832, 2019.

Michelle Stephens, Nelson Turner, and Jon Sandberg, "Particle identification by laser-induced incandescence in a solid-state laser cavity," Appl. Opt. 42, 3726-3736 (2003)

---

## Referee Comment (RC2) · J. C. Corbin (Referee) · 24 Nov 2020

The manuscript amt-2020-416 by Howell et al. presents a systematic investigation into the reasons that an infrared-laser optical particle sizer, the UHSAS, could have reported optical particle diameters (Dopt) at m=1.572+0i smaller than mobility particle diameters (Dmob) during an aircraft study of biomass-burning plumes. This is essentially an attempt at closure between Dopt and Dmob for particles of potentially complex morphology and refractive index, and is a difficult enough task even without considering particle evaporation in the UHSAS.

The authors support the interpretation of their field data using laboratory experiments,

Mie-theory (spherical-particle) calculations, and particle-heating calculations. The interpretation is presented as a series of hypotheses which are eliminated one by one, which is a particularly clear approach. The main conclusions are that the UHSAS measures (1) Dopt ≈ Dmob for non-absorbing spheres, (2) Dopt ≈ 0.9*Dmob for the measured brown carbon, and (3) Dopt ≤ 0.7*Dmob for anomalously low scatterers (black carbon or tarballs). The authors attribute Conclusion (2) to partial evaporation and therefore propose (4) an empirical correction to force Dopt' = 1*Dmob for their brown carbon samples.

Overall, the work is of excellent quality and contributes substantially to the understanding of IR laser particle sizers (UHSAS) as well as including some conclusions applicable to the single particle soot photometer (SP2). I have some major concerns, but these can be practically addressed by adding a few calibration particle types. So I will request that these experiments are included in a revised manuscript.

**Major comments**

Briefly, my two major concerns are (i) an alternative hypothesis for the field data is internal mixing of soot and organic matter (OM), and (ii) the UHSAS has a similar laser intensity than the SP2. The second concern implies that tarballs should evaporate in the UHSAS (in support of this work's conclusions), rBC should be vapourized and not detected (in apparent contradiction of the laboratory work), and particle heating is much larger than currently calculated, at least for soot.

These two concerns are fleshed out in the following comments, which address the manuscript more directly.

1. The evaporation hypothesis has not been unambiguously shown from the laboratory experiments. The authors have not measured absorbing spheres to demonstrate evaporation. It would be relatively simple for the authors to reproduce the experiments of Sedlacek et al. (2018), using nigrosin. This would let the authors clearly demonstrate what the hypothesized evaporation effects would look like in the UHSAS. Ideally, different laser powers would be used (by varying the pump laser power). Because evaporation in the UHSAS is a strong claim, it should be supported by this direct demonstration.

2. The laboratory experiments have not shown the response of the UHSAS to realistic soot with a DLCA morphology. This morphology plays a major role in the light scattering properties of atmospheric black carbon (Sorensen et al., 2018). Light scattering by fractal soot aggregates is significantly lower than that of equivalent spheres due to its morphology alone (Mishchenko, 2009's Figure 12). Atmospheric black carbon either has DLCA morphology or is compacted from DLCA by coatings. The fullerene soot sample may at best represent compacted DLCA soot, which is adequate but could be much improved by a simple experiment with a kerosene flame or similar.

3. The calibration experiments are also missing a non-absorbing, non-spherical case, which would help to explain whether absorption is really important here, or just morphology. Silica or titania aggregates could be used (Schmoll et al., 2009) and would ideally be generated as DLCA aggregates (Eggersdorfer and Pratsinis, 2013) for comparison with soot. The authors may avoid this suggestion by including both soot and absorbing spheres, however.

4. In the scattering calculations corresponding to the calibrations, the authors should follow the literature to use the relatively simple RDGFA approach (Sorensen et al., 2018) rather than an effective medium approach in approximating soot properties. The combustion literature has long used the RDGFA approach to obtain reasonable results for soot and to show that soot scatters very little light at similar wavelengths (Liu et al., 2019). With the addition of an

absorbing spherical particle type, a DLCA soot sample, and optionally a non-absorbing DLCA aggregate, the authors' work would represent a comprehensive study of the UHSAS response. So, these first 4 comments would not only close an important gap in the reasoning here but provide valuable reference data for others.

5. I have requested a direct demonstration of the evaporation hypothesis because I can propose an alternative hypothesis which the authors have not discussed: internal mixing of soot with non-absorbing or slightly absorbing material. Internal mixing is almost inevitable for plumes as old as those studied here (2 days to 2 weeks, Line 309). The authors' laboratory data shows that black-carbon surrogates scatter much less than predicted by equivalent spheres (as expected and noted in the previous comment). It can be expected that coated black carbon would behave somewhere in between soot and non-absorbing spheres, as suggested by previous work (Mikhailov et al., 2006). This does not univocally imply a continuous range of UHSAS signals in Figure 7, because particle breakup due to laser heating (Moteki and Kondo, 2007) could cause two UHSAS modes: either coatings evaporate and give a smaller signal, or they fragment and give a larger signal. The field data (authors' Figure 7) can therefore be explained simply as a mixture of soot, organic matter (OM), and soot+OM particles:

   (a) The soot particles are the anomalous low scatterers with $D_{mob} > 100$nm, as expected for DLCA aggregates.

   (b) The OM particles are the smallest particles (circles and squares with $D_{mob} < 100$nm).

   (c) The soot+OM particles are the larger particles (circles and squares with $D_{mob} > 100$nm).

   (d) The ratio of $D_{opt}/D_{mob}$ (y axis of Figure 7) may decrease with increasing $D_{mob}$ because larger soot particles have larger shape factors (Sorensen,

2011), or because larger soot particles have larger internal coupling parameters (Sorensen et al., 2018), or both. I made a rough calculation of the latter effect and it appears to be smaller than the former.

A second internal-mixing hypothesis replaces soot with tarballs above.

The current laboratory experiments on soot surrogates (Aquadag and fullerene soot) actually support this internal-mixing hypothesis more than the brown-carbon evaporation hypothesis. These surrogates anyway probably scatter more light than soot since their structure is more compact, and since Aquadag is made up of larger graphite flakes. The authors may perhaps consider my alternative as a sub-set of their evaporation hypothesis, or disprove it using their thermal denuder data.

6. In the context of the previous comments, I question the value of a "correction" to the UHSAS. If the surpising signals represent real physical phenomena, and the UHSAS is working correctly, why "correct" the data? Section 3.2 could instead follow the tone of Section 3.1, and focus on the prediction of the UHSAS response from fundamental particle properties. Since the particle properties are not known exactly, the properties (morphology effects on scattering, morphology effects on Dmob, refractive index and – if justified – evaporated volume fraction) required to explain the observations can be discussed. If the authors' answer is that a correction is valuable to predict volumetric size distributions and total light scattering, then please modify the manuscript to emphasize this.

7. The manuscript has cited relevant SP2 work but there are a few points where the SP2 literature should be used to constrain the UHSAS predictions.

   (a) The intensity of the SP2 laser has been reported as $1.7E+05$ $W\,cm^{-2}$, $6.5E+05$ $W\,cm^{-2}$, and $4.05E+05$ $W\,cm{-2}$ by Schwarz et al., 2006, Moteki and Kondo 2007, and Bambha and Michelsen 2015, respectively. Cai et al.

   (2008) reported the UHSAS laser intensity as $5.1E+05$ $W\,m{-2}$. These are all similar to the $5.1E+09$ $W\,m^{-2}$ ($5.1E+05$ $W\,cm^{-2}$) reported here. The SP2 and UHSAS wavelengths are about the same.
   So, the behaviour of a given particle in the SP2 can be extrapolated to the UHSAS. (While thinking about this I consulted Figure 1 of Corbin and Gysel-Beer 2019, which shows the behaviour of various particle types in the SP2 laser.) Any particles which vapourize (or carbonize) in the SP2 must vapourize in the UHSAS. Therefore, Equation 1 and Figure 4 cannot be correct. The SP2 routinely observes soot particles down to about 80 nm from their incandescence at >3000 K, yet Equation 1 apparently predicts only 1000 K at steady state for 100 nm soot. Moreover, this size is an overestimation for aggregate particles like fullerene soot. The assumptions behind Equation 1 must be flawed, at least for non-spherical particles. Bambha and Michelsen (2015) performed more detailed calculations than Cai et al. (2008); probably too detailed for this manuscript. So it becomes even more important to perform UHSAS calibration experiments with DLCA soot.

   (b) Sedlacek et al. (2018) showed that nigrosin (brown carbon / tarball surrogate) absorbs the SP2 laser. Corbin and Gysel-Beer (2019) reported SP2 time-resolve scattering cross-sections for the evaporation of tar brown carbon (TB) from heavy-fuel oil that was similar to Alexander et al. (2008)'s tarballs Their scattering cross-sections for TB actually look very similar to the "anomalous" particles reported here. In contrast, their scattering cross-sections for rBC actually show that the rBC evaporates before reaching the centre of the laser.
   So, how can the UHSAS see rBC? Presumably, the software uses the peak signal for all particles. This would correspond to the peak laser intensity for non-evaporating particles, but would occur before the peak for evaporating particles. This difference in incident laser intensity would result in an under-sizing of evaporating particles, in addition to their actual change in volume.

(I have presumed that the software uses the peak signal because the SP2 would have to do the same if it did not have a "split" detector.) This is an important issue which will be implicitly addressed by the addition of soot particles in the calibrations.

**Minor comments**

While reading the manuscript carefully a number of minor comments arose, which I list here.

1. I appreciated the original structure of the introduction, but please add a short goals paragraph at the end. Please also consider moving parts of Section 1.3 to Methods.

2. Please add an Appendix section where the prediction of nephelometer signals from UHSAS signals is explicitly described. I can imagine what was done but it should be spelt out.

3. In Section 1.2 of the introduction, the authors' use of light-absorbing carbon (LAC) terminology could be refined. The section should cite Petzold et al. (2013) in its first paragraph, which is a review of the topics discussed there.

   In the next paragraph, change boiling to incandescence (line 79) since rBC actually sublimates.

   Please reword the statement that BC is a hypothetical material that includes graphitic soot nanospheres and amorphous C, citing either the definitions given by Petzold et al. (2013), Bond and Bergstrom (2006). The word hypothetical suggests that soot is not of a consistent composition, which may mislead readers (Michelsen et al. 2020).

   Please also reconsider the statement that quantitatively connecting the amount of carbon with the light absorption of LAC remains a challenge. The issue is not that connecting the two is challenging, but that the range of light-absorbing compounds formed by carbon is vast. These LAC compounds include brC, black carbon, and amorphous carbon in tarballs (a recent summary is given in Corbin et al., 2019). When the authors mention amorphous C, are they referring to

the degree of graphitization (Michelsen et al., 2020) of the sample? The term amorphous C is often used for a specific carbon material, rather than amorphous domains within soot. Please reword to clarify.

4. Please add a short description of the UHSAS to Methods. How are particle focussed into the beam, at what flow rate, etc.

5. Please mention the particle counter after the DMA in Methods. Of course, a DMA by itself does not provide size distributions, so I assume a CPC was used. Similarly, at line 126, please clarify that it is not the DMA but the CPC which cannot tell the difference between charges (especially important since this work uses a UHSAS to tell the difference).

6. Line 147 mentions an empirical correction to a valve – please clarify if this correction was applied to CPC, UHSAS, or both. Please consider adding a graph to the supplement and/or stating here the magnitude of the correction.

7. Line 151 and surrounding, please mention what RH the sample would have had without dessication or what the maximum dewpoint would have been (I assume it is very low given the altitude).

8. Line 161 please briefly mention the reason why the fraction is trivial, especially since the detector is missing from Figure 1. I am assuming that the reason is the small collection angle of the detector optics.

9. Figure 2 please change soot to ambient soot with a citation to Moteki et al (2010) in the legend. I originally misunderstood it as fullerene soot. Please also change graphite to graphite sphere (or similar) for clarity, since Aquadag is also graphite.

10. Line 180 please provide a citation for this description of fullerene soot. Please change to the RDG approximation here.

11. Line 191 "poor statistics"... how exactly was the statistical analysis done? Was the mode or median of the distributions used?

12. Line 192 please change "than expected" to "than predicted for equivalent spheres" or similar.

13. Table 1, what is "amorphous C" here? I did not see this term used by Moteki et al. 2010. They did use the term "non-graphitic" to describe samples with a lower degree of graphitization. I suggest omitting this entirely as the optical properties of soot (one row above) will be similar. This comment relates to my general comment on LAC terminology above.

14. Table 1, please indicate either how these values were extrapolated from measurements at other wavelengths, or the measurement wavelength if they were not.

15. Table 1, please change 'variable' to the values used in order to convey more precise information.

16. Line 124, I do not agree that Gysel et al. (2011) concluded that Aquadag particles are 13% unknown composition after denuding at 450 degrees C, and I do not see the statement in that work. These particles would be 100% rBC by definition, but may have a different SP2 response to other forms of rBC. In other words, this statement is illogical since the SP2 is calibrated to the total mass of denuded particles. Perhaps the authors are referring here to the EC content of Aquadag; this excludes oxygen and other atoms, so is smaller than rBC mass.

17. Figure 6 should not show the extrapolation if the discussion describes it as "completely inappropriate", please harmonize.

18. Line 250 please change "kernel function" to "transfer function" according to convention.

19. Line 303 please report the wavelengths which the AAE was calculated from.

20. Throughout the manuscript, error bars were generally missing and not discussed, please add them or an overall comment.

21. Line 204 please mention $1/e^2$ as the measure of beam diameter (if correct).

22. Line 382 much lower temperatures than 400 C, are enough to evaporate most coatings. e.g. https://doi.org/10.1016/j.jhazmat.2011.12.061

**References**

Corbin, J. C. and Gysel-Beer, M.: Detection of tar brown carbon with a single particle soot photometer (SP2), Atmos. Chem. Phys., 19(24), 15673–15690, doi:10.5194/acp-19-15673-2019, 2019.

Corbin, J. C., Czech, H., Massabò, D., de Mongeot, F. B., Jakobi, G., Liu, F., Lobo, P., Mennucci, C., Mensah, A. A., Orasche, J., Pieber, S. M., Prévôt, A. S. H., Stengel, B., Tay, L. L., Zanatta, M., Zimmermann, R., El Haddad, I. and Gysel, M.: Infrared-absorbing carbonaceous tar can dominate light absorption by marine-engine exhaust, npj Clim. Atmos. Sci., 2(1), 12, doi:10.1038/s41612-019-0069-5, 2019.

Eggersdorfer, M. L. and Pratsinis, S. E.: Agglomerates and aggregates of nanoparticles made in the gas phase, Adv. Powder Technol., 2013.

Liu, F., Yon, J., Fuentes, A., Lobo, P., Smallwood, G. J. and Corbin, J. C.: Review of recent literature on the light absorption properties of black carbon: Refractive index, mass absorption cross section, and absorption function, Aerosol Sci. Technol., 54(1), 33–51, doi:10.1080/02786826.2019.1676878, 2020.

Michelsen, H. A., Colket, M. B., Bengtsson, P.-E., D'Anna, A., Desgroux, P., Haynes, B. S., Miller, J. H., Nathan, G. J., Pitsch, H. and Wang, H.: A Review of Terminology Used

to Describe Soot Formation and Evolution under Combustion and Pyrolytic Conditions, ACS Nano, 14(10), 12470–12490, doi:10.1021/acsnano.0c06226, 2020.

Mikhailov, E. F., Vlasenko, S. S., Podgorny, I. A., Ramanathan, V. and Corrigan, C. E.: Optical properties of soot–water drop agglomerates: An experimental study, J. Geophys. Res., 111(D7), D07209, doi:10.1029/2005JD006389, 2006.

Mishchenko, M. I.: Electromagnetic scattering by nonspherical particles: A tutorial review, J. Quant. Spectrosc. Radiat. Transf., 110(11), 808–832, doi:10.1016/j.jqsrt.2008.12.005, 2009.

Petzold, A., Ogren, J. A., Fiebig, M., Laj, P., Li, S.-M., Baltensperger, U., Holzer-Popp, T., Kinne, S., Pappalardo, G., Sugimoto, N., Wehrli, C., Wiedensohler, A. and Zhang, X.-Y.: Recommendations for the interpretation of "black carbon" measurements, Atmos. Chem. Phys., 13(16), 8365–8379, doi:10.5194/acp-13-8365-2013, 2013.

Schwarz, J. P., Gao, R. S., Fahey, D. W., Thomson, D. S., Watts, L. A., Wilson, J. C., Reeves, J. M., Darbeheshti, M., Baumgardner, D. G., Kok, G. L., Chung, S. H., Schulz, M., Hendricks, J., Lauer, A., Kärcher, B., Slowik, J. G., Rosenlof, K. H., Thompson, T. L., Langford, A. O., Loewenstein, M. and Aikin, K. C.: Single-particle measurements of midlatitude black carbon and light-scattering aerosols from the boundary layer to the lower stratosphere, J. Geophys. Res., 111, D16207, doi:10.1029/2006JD007076, 2006. Schmoll, L. H., Elzey, S., Grassian, V. H. and O'Shaughnessy, P. T.: Nanoparticle aerosol generation methods from bulk powders for inhalation exposure studies, Nanotoxicology, 3(4), 265–275, doi:10.3109/17435390903121931, 2009.

Sedlacek, A. J., Onasch, T. B., Nichman, L., Lewis, E. R., Davidovits, P., Freedman, A. and Williams, L.: Formation of refractory black carbon by SP2-induced charring of organic aerosol, Aerosol Sci. Technol., 52(12), 1345–1350, doi:10.1080/02786826.2018.1531107, 2018.

Sorensen, C. M.: The mobility of fractal aggregates: a review, Aerosol Sci. Technol.,

45(7), 765–779, doi:10.1080/02786826.2011.560909, 2011.

Sorensen, C. M., Yon, J., Liu, F., Maughan, J., Heinson, W. R. and Berg, M. J.: Light scattering and absorption by fractal aggregates including soot, J. Quant. Spectrosc. Radiat. Transf., 217, 459–473, doi:10.1016/j.jqsrt.2018.05.016, 2018.

---

## Short Comment (SC1) · 21 Dec 2020

This is a very interesting and well written paper on the UHSAS sizing behavior for absorbing aerosols encountered in an aged biomass burning plume. The topic overlaps with some recent laboratory work that we've been doing at NASA Langley to understand the performance of our Laser Aerosol Spectrometer (LAS) and UHSAS instruments, as well as the airborne measurements of fresh biomass burning plumes conducted during the 2019 FIREX-AQ field campaign. In this paper, Howell et al. use combined electrical mobility and optical particle sizing to study the UHSAS response to biomass burning particles. The main findings are reported as

- Particles with electrical mobility diameters between 70 and 280 nm are optically sized by the UHSAS, and the singly-charged monodisperse aerosols between 100-200nm diameters show up as two peaks on the UHSAS.

- The peak with 70-100% of the particle number has a mode diameter that is undersized relative to the mobility diameter set point by 0-15% (depending on size), and Mie theory calculations show that particle composition-dependent refractive index changes are unlikely to explain this undersizing.

- The other "anomalous" peak with 5-30% of particles has a mode diameter that is even more significantly undersized by 25-35%, which is consistent with the UHSAS undersizing of laboratory-generated, fullerene soot particles.

- The ORACLES polydisperse size distributions are corrected using a power-law fit to the monodisperse data between 70-280nm that is extrapolated to 600 nm, while Mie theory is used for particles larger than 600 nm.

- The reader is cautioned that "UHSAS data should be treated cautiously whenever the aerosol may absorb infrared light".

**In this short comment, I ask that the authors consider the following:**

1. The main peak undersizing of 0-15% (depending on size) is largely consistent with the difference in UHSAS size response between PSL particles and size-classified ammonium sulfate particles, and that the finding reported in the second bullet point above is due to differences in refractive index between the PSL calibration standard and aged, biomass burning aerosols.

2. Can the dual peaks between 100-180 nm be attributed to a stitching error in the transition region between the G3 to G2 gain stages, where the smaller 'anomalous' peak is from the G3 detector? While such a stitching error would not bias

the polydisperse size distribution (or indeed even be noticeable in many cases), it may give rise to extra peaks when looking at monodisperse aerosols near the gain stage transition point. What was the G3 gain stage saturation diameter for this instrument during the ORACLES campaigns as well as the subsequent tests?

3. Differences between Mie theory and the actual instrument performance may be substantial between 600-1000 nm as indicated by Figure 3, which may explain the poor performance of the extrapolated correction. The difference between NaCl (refractive index of 1.53+0i) and PSLs is particularly noticeable and seems to exhibit 20% undersizing between 600-1000 nm.

4. Along the lines of my Point 1 and the authors' caution in the final bullet above, it's not clear to me that aged biomass burning particles or other atmospherically-relevant absorbing aerosols far from emissions sources are meaningfully different from non-absorbing aerosols in terms of UHSAS sizing. Instead, these results motivate the need to calibrate the UHSAS with particles of atmospherically-relevant refractive index instead of PSLs.

**Discussion:**

Figure SC1 is included below to support these considerations, which shows mobility-classified ammonium sulfate (AS) sizing data for our UHSAS. We have used these data in past field campaigns to convert PSL-calibrated size bins to AS-calibrated size bins (e.g., Sawamura et al., 2017). Ammonium sulfate was chosen as the calibrant following Brock et al., 2011, and its real refractive index (1.52) is within the range of 1.52-1.54 identified by Shingler et al. (2016) as being representative of average aerosols encountered during SEAC4RS over a diverse range of air mass types (urban, marine, biogenic, and biomass burning). The

biomass burning peak sizes are consistent with the ammonium sulfate curve up to about 200nm, after which the curves diverge slightly. It seems reasonable to expect that this might be caused by slight differences in the shape of the internal instrument calibration curves, and **I'd suggest that the authors compare their biomass burning curve from Figure 6 to the NaCl curve from Figures 2-3 rather than relying only on Mie Theory calculations as in Figure 7 to rule out the refractive index explanation.** Figure 3 shows that the Mie calculations significantly overestimate the expected size response for NaCl at all sizes. The refractive index of NaCl (1.53+0i) is very close to that of AS, and our experiments and those reported by Cai et al. (2008) indicate that the NaCl asphericity has a negligible effect on the UHSAS sizing of DMA-classified aerosols.

I find the presence of the 'anomalous' peak very interesting, and I enjoyed reading the authors' well-written discussion of potential artifacts caused by particle absorption and heating (and the limitations of this explanation given theoretical temperature increases). The laboratory measurements clearly show substantial undersizing of the absorbing species, and this is consistent with Kupc et al. (2018) as well as our past laboratory work (Figure SC2), which I've included here in case it helps inform the conclusions related to the difference between fullerene soot and nigrosine dye sizing. That the 'anomalous' particle size curve lines up so well with that for fullerene soot seems like more than a coincidence and would seem to imply the existence of a small, externally-mixed aerosol population in this aged biomass burning plume.

An alternative explanation that I'd like to hear the authors' thoughts on is that that the anomalous peak is due to a slight gain stage stitching error at the G3-G2 transition that causes a small fraction of particles to be sized low by the G3 detector up to its saturation limit. These sorts of stitching errors seem to be fairly common in our measurements but are generally inconsequential given

the noise in a the polydisperse size distributions. However, for a monodisperse aerosol, this might result in the small side peak that would be misinterpreted as an externally-mixed aerosol mode. As an example, I've included the gain stage saturation points for our UHSAS during a recent PSL calibration in Figure SC1, but I'd expect that these sizes would be different for the authors' instrument and calibration during ORACLES. **In sum, can a gain stage stitching error be ruled out as the explanation for the 'anomalous' peak?**

**References:**

Cai, Y., Montague, D. C., Mooiweer-Bryan, W., and Deshler, T.: Performance characteristics of the ultra high sensitivity aerosol spectrometer for particles between 55 and 800nm: Laboratory and field studies, Journal of Aerosol Science, 39, 759-769, https://doi.org/10.1016/j.jaerosci.2008.04.007, 2008.

Kupc, A., Williamson, C., Wagner, N. L., Richardson, M., and Brock, C. A.: Modification, calibration, and performance of the Ultra-High Sensitivity Aerosol Spectrometer for particle size distribution and volatility measurements during the Atmospheric Tomography Mission (ATom) airborne campaign, Atmos. Meas. Tech., 11, 369-383, 10.5194/amt-11-369-2018, 2018.

Sawamura, P., Moore, R. H., Burton, S. P., Chemyakin, E., Müller, D., Kolgotin, A., Ferrare, R. A., Hostetler, C. A., Ziemba, L. D., Beyersdorf, A. J., and Anderson, B. E.: HSRL-2 aerosol optical measurements and microphysical retrievals vs. airborne in situ measurements during DISCOVER-AQ 2013: an intercomparison study, Atmospheric Chemistry and Physics, 17, 7229-7243, 10.5194/acp-17-7229-2017, 2017.

Shingler, T., Crosbie, E., Ortega, A., Shiraiwa, M., Zuend, A., Beyersdorf, A., Ziemba, L., Anderson, B., Thornhill, L., Perring, A. E., Schwarz, J. P., Campazano-Jost, P., Day, D. A., Jimenez, J. L., Hair, J. W., Mikoviny, T., Wisthaler, A., and Sorooshian, A.: Airborne characterization of subsaturated aerosol hygroscopicity and dry refractive index from the surface to 6.5 km during the SEAC4RS campaign, Journal of Geophysical Research: Atmospheres, 121, 4188-4210, https://doi.org/10.1002/2015JD024498, 2016.

Zimmerman, S., Moore, R., Anderson, B., Beyersdorf, A., Corr, C., Shook, M., Thornhill, K., Winstead, E., and Ziemba, L.: Evaluation of Combined Electrical Mobility and Optical Sizing Techniques for Deriving Aerosol Refractive Index, American Association for Aerosol Research (AAAR) Annual Meeting, Minneapolis, MN, USA, 2015.

[Figure]

**Fig. 1.** UHSAS optical diameters vs. DMA electrical mobility diameters for digitized data pts. in the present paper as well as NASA ammonium sulfate calibrations. Gain stage transition pts. are also shown.

[Figure]

**Fig. 2.** UHSAS size response to aerosol of different chemical compounds with reported refractive indices from the literature. Reproduced from Zimmerman et al. (2015).

---

## Author Comment (AC1) · 13 Sep 2021

**Response to comments on "Undersizing of Aged African Biomass Burning Aerosol by an Ultra High Sensitivity Aerosol Spectrometer" by Charles Brock**

Steven Howell and Steffen Freitag

September 12, 2021

We'd like to thank Charles Brock for his review. His comments point out ways to make the paper clearer, and identified the single biggest weakness–that the UHSAS laser intensity is actually comparable to the SP2, so incandescence and particle vaporization are likely to happen. It turns out that the heat transfer calculation we used to demonstrate that incandescence was unlikely was in fact inappropriate, as it assumed that particles were much larger than the mean free path of air molecules. This realization had substantially changes our conclusions.

**Comments from Charles Brock**

1. *Line 66. You may want to cite Kupc et al. (2018) here. Kupc et al. describe modification and calibration of a UHSAS for airborne use, and is one of only two (now three) papers discussing the performance of the UHSAS.*

   I'm sorry to say I had missed that paper. It's a good one! It's now cited.

2. *Line 106. You mention later (line 139) that a "grab" sampler was used by the SMPS. Why do you need a "vast" and apparently homogeneous plume to do the size-resolved analysis? Doesn't the grab sample eliminate the need for homogeneity?*

   That is a good point–this kind of in-flight calibration can be done at a small scale, but there are a couple of limitations. Most obviously, the grab sampler only has enough volume for 3 scans, so multiple grabs were necessary to characterize the UHSAS response. More importantly, if aerosol composition changes at small scales, then one cannot assume that a calibration at one location is valid elsewhere in the plume. The consistency of our results over many samples and suggest that they are generally valid for this large plume.

3. *Figures 2-5. These figures are well laid out, but they use very similar colors and line types. The colors for the graphite, Aquadag, and NaCl are too similar, as are the sulfate types. I'm not color-vision-impaired, but I know many who are, and these figures would be really tough to read. Can you use more distinctive colors and different line types (dotted, dashed, dash-dotted, etc.) to better distinguish the curves?*

   I have tried to make those figures clearer, altering line types and choosing colors that shouldn't be confusing to the most common types of colorblindness. Part of the difficulty is that the symbols are overloaded–the colors identify test materials and the shapes indicate the number of charges. In my opinion, the number of charges information information is worthwhile, but it could be excluded, allowing the symbol shapes to clearly identify the test materials.

4. *Line 236. You state that the SP2 has a "much more powerful IR laser". Stephens et al., (2003), who describe the prototype for the SP2, quote a laser intensity of $10^6\,\mathrm{W\,cm^{-2}}$. This is not very different from the stated (line 204) UHSAS laser intensity of $\sim5\times10^9\,\mathrm{W\,m^{-2}}$ ($= 0.5\times10^6\,\mathrm{W\,cm^{-2}}$). Thus I*

*would not be surprised at all if the UHSAS is able to at least partially incandesce BC particles, leading to a mis-sizing.*

This turns out to be an extremely important point. The SP2 laser system is so much more elaborate that it hadn't occurred to me that the energy density is comparable. As it turns out, the equations I used for thermal conduction to the air, following Cai et al. (2008), which assume particles are large enough to be in the continuum regime, indicate that the SP2 would be unable to heat small rBC particles to incandescence. That's clearly wrong! It turns out that applying a transitional regime model, as in Bambha and Michelsen (2015), makes a huge difference. It is now clear that incandescence and vaporization is a very real problem in the UHSAS. The paper now reflects that.

5. *Line 357. The corrected UHSAS data are not "an order of magnitude too high near* $1 \, \mu m$*" for the free troposphere case (Fig. 8f). It's less than a factor of 2.*

Good point. All of the corrected UHSAS volume distributions are high, but only the MBL example is high by nearly an order of magnitude. I've rephrased it to "the corrected UHSAS is considerably higher than the APS between 0.6 and $1 \, \mu m$"

6. *Line 373. Here you say that it's not clear which refractive index to use to calculate scattering for comparison with the TSI nephelometer. Each bin of the UHSAS corresponds to a certain amount of scattering into the detection volume. If you quantify how much scattering each bin represents, which you are effectively doing by calibrating with a monodisperse aerosol, you should just use the same calibrant refractive index to calculate total scattering. In effect, you are just summing up the scattering represented by each bin, getting the total scattering. Of course, this ignores the difference between hemispherically integrated scattering vs. the narrower viewing angles of the UHSAS detection optics, but it is a very good first approximation to just use the refractive index of the calibrant (in this case, PSL). You can investigate the magnitude of the error due to the scattering geometry using Mie calculations.*

That's what we've normally done in the past and in fact what we have done here. However, the situation is made more complicated by the fact that the refractive indices at the UHSAS wavelength are different than those in the visible, so the accuracy of the results are affected by the relative change in refractive index of the sample particles and the calibration spheres. I've explored that a bit in Fig 1.

7. *Line 371. Can you use the rBC number fraction from the SP2 to estimate the number of anomalously undersized particles, and boost the number in the main mode by this fraction to compensate?*

Certainly one could do that, but that would require making assumptions about exactly which size particles to move.

8. *Line 384. The obvious explanation is that the coatings are not volatile at 400 C. This is consistent with Adler et al., 2019, who found coatings on biomass burning particles that did not evaporate (at lower temperature) but that were not incandescent in the SP2.*

The new heat loss calculations indicate that even a pretty small amount of absorption by rBC can heat up particles sufficiently to volatilize almost anything organic, so this isn't really an issue any more. I expect that the coating simply takes long enough to volatilize that the peak scattering occurs at the initial peak as shown in Fig 1c of Laborde et al. (2012) rather than at the second peak.

9. *Fig. 9. Have you modified the UHSAS flow system as in Kupc et al.? We found that both of our UHSAS instruments leaked through the seals around the detectors, downstream of the detection region, reducing the sample flow even though the exhaust flow (which is the nominal flow measurement) was constant. This produced an altitude-dependent flow bias (Brock et al. https://doi.org/10.5194/acp-11-2423-2011), although it works in the direction opposite the trend seen here. Droplet Measurement Technologies has repaired the leak in our UHSAS, and they had a jig and setup to do this, implying that this is a common problem that they have had to fix in the past.*

No. Unfortunately, our field project was over before the paper was published. We actually did notice a small discrepancy between inlet and outlet flow rates but failed to find the leak. It was small enough that we did not notice an altitude dependence, although we looked for it.

10. *Fig. 11. Change y-axis label to "Fraction of Particles".*

    That's not actually what the Y axis is. The x-axis is the fraction of rBC-containing particles. The y-axis is the number of sample periods that had that fraction of rBC-containing particles normalized by the total number of samples each year. The sample periods were rather arbitrarily chosen as LDMA scan periods, which were 85 s long. I've clarified that in the text.

11. *The Appendix is very clear and helpful.*

    Thanks! I hope the two more appendices are useful as well.

12. *Please check over the references for consistency with Copernicus formatting guidelines. For example, Clarke and Ellis et al. have capitalized titles, journal names are not consistent, etc. This is a consequence of EndNote-type software, which ALWAYS needs manual checking and correction.*

    Yes, something always slips through. I'm actually using the rather outdated LaTeX and BibTeX style files supplied by Copernicus, so EndNote can't be blamed this time.

**References**

Bambha, R. P. and H. A. Michelsen (2015). "Effects of aggregate morphology and size on laser-induced incandescence and scattering from black carbon (mature soot)". *J. Aerosol Sci.* 88, pp. 159–181. DOI: 10.1016/j.jaerosci.2015.06.006.

Bowen, N. L. (1926). "Properties of Ammonium Nitrate. I". *The Journal of Physical Chemistry* 30.6, pp. 721–725. DOI: 10.1021/j150264a001.

Cai, Y., D. C. Montague, W. Mooiweer-Bryan, and T. Deshler (2008). "Performance characteristics of the ultra high sensitivity aerosol spectrometer for particles between 55 and 800nm: Laboratory and field studies". *J. Aerosol Sci.* 39.9, pp. 759–769. DOI: 10.1016/j.jaerosci.2008.04.007.

Laborde, M., P. Mertes, P. Zieger, J. Dommen, U. Baltensperger, and M. Gysel (2012). "Sensitivity of the Single Particle Soot Photometer to different black carbon types". *Atmos. Meas. Tech.* 5.5, pp. 1031–1043. DOI: 10.5194/amt-5-1031-2012.

[Figure]

Figure 1: Effect of refractive index on calculated scattering. Panel a) shows the power scattered into the UHSAS optics for 3 materials. The PSL line is almost precisely on top of the $NH_4NO_3$ since the refractive indexes are almost identical. I then determined how big the $NH_4NO_3$ and $H_2SO_4$ particles would have to be to be classified as a PSL particle of a given size. Panel b) shows the ratio of those sizes. When calculating scattering per particle at the UHSAS wavelength (panel c), one sees good agreement up to $0.4\,\mu m$ for both salts, but then the large size of the $H_2SO_4$ particles does a lot of forward scattering that isn't seen by the UHSAS. Nevertheless, one would expect very good scattering closure for the accumulation mode. However, when calculating scattering in the mid-visible, the refractive indices of the materials change in different ways, yielding much poorer agreement, something like a 10% underestimate for the accumulation mode. $NH_4NO_3$ refractive indices for visible light are averages of the 3 axes of orthorhombic crystals (the stable form from $-16\,°C$ to $31.2\,°C$) from Bowen (1926) and for infrared light are from Cai et al. (2008).

---

## Author Comment (AC2) · 22 Sep 2021

**Response to comments on "Undersizing of Aged African Biomass Burning Aerosol by an Ultra High Sensitivity Aerosol Spectrometer" by J. C. Corbin**

Steven Howell and Steffen Freitag

September 21, 2021

We'd like to thank Dr. Corbin for his review. He clearly took great care and pointed us towards real problems with some of our calculations. It will disappoint him that we are not actually in a position to run significant experiments with the UHSAS, as it needs repair for which we don't have funds at the moment. We hope to perform the tests he recommended at some point in the near future (or read about someone else doing it). So we don't have the thorough theoretical plus experimental analysis of the UHSAS that he would like to see. However, we have done much of the reanalysis that he requests and hope this revised version will be regarded as a significant enough contribution to the literature.

The bottom line is that Dr. Corbin was entirely correct in his suspicion of Eq. 1, which applies only in the continuum regime, when the mean free path $L$ is much smaller than particle diameter $D_p$. While the particles were indeed smaller than $L$, the non-continuum effects are significant even when $D_p > 20L$. When implementing the transitional scheme of McCoy and Cha (1974) as used in Bambha and Michelsen (2015), it is clear that our previous estimates of particle heating were wild underestimates. There are significant uncertainties, but it is clear that raising rBC particles to volatilization temperatures is exceedingly likely. That resolves many of the issues we had trouble with and changes many of the arguments we made.

I also implemented more realistic optical calculations, though not the Rayleigh-Gans-Debye approximation recommended. Instead, I used Multiple Sphere $T$-matrix (MSTM v3.0) (Mackowski, 2014; Mackowski and Mishchenko, 2011), which requires similar assumptions about the particles (nonintersecting spheres) but is a numerical solution rather than an approximation, and is more accurate for absorption calculations (Mackowski, 2006; Sorensen et al., 2018). It wound up not making a large difference in particle sizing, but added roughly 10 % to the absorption and hence particle heating. The way MSTM was used is detailed an a new appendix.

In the rest of this document, Dr. Corbin's comments are in **bold**.

**Major concerns**

**Briefly, my two major concerns are (i) an alternative hypothesis for the field data is internal mixing of soot and organic matter (OM), and (ii) the UHSAS has a similar laser intensity than the SP2. The second concern implies that tarballs should evaporate in the UHSAS (in support of this work's conclusions), rBC should be vapourized and not detected (in apparent contradiction of the laboratory work), and particle heating is much larger than currently calculated, at least for soot. These two concerns are fleshed out in the following comments, which address the manuscript more directly.**

The idea that rBC would not be detected because it was vaporized seems incorrect, because the particle must enter the beam to get heated. Once in the beam it will scatter light and be detected. As the particle progresses father into the beam, two processes compete: scattering gets stronger as the light intensity increases, and scattering is reduced as the particle vaporizes.

**1. The evaporation hypothesis has not been unambiguously shown from the laboratory experiments. The authors have not measured absorbing spheres to demonstrate evaporation. It would be relatively simple for the authors to reproduce the experiments of Sedlacek et al.**

(2018), using nigrosin. **This would let the authors clearly demonstrate what the hypothesized evaporation effects would look like in the UHSAS. Ideally, different laser powers would be used (by varying the pump laser power). Because evaporation in the UHSAS is a strong claim, it should be supported by this direct demonstration.**

As mentioned above, we are not able to do that experiment at the moment. It would be a really good idea. However, in light of the new calculations of particle heating, there is very little reason to doubt that evaporation would occur.

**2. The laboratory experiments have not shown the response of the UHSAS to realistic soot with a DLCA morphology. This morphology plays a major role in the light scattering properties of atmospheric black carbon (Sorensen et al., 2018). Light scattering by fractal soot aggregates is significantly lower than that of equivalent spheres due to its morphology alone (Mishchenko, 2009's Figure 12). Atmospheric black carbon either has DLCA morphology or is compacted from DLCA by coatings. The fullerene soot sample may at best represent compacted DLCA soot, which is adequate but could be much improved by a simple experiment with a kerosene flame or similar.**

We do actually have some lab data of particles generated by a kerosene lamp (Fig. 1), but omitted it from the paper because

1. freshly generated kerosene soot was not representative of the types of aerosols seen in ORACLES;

2. we do not know how much rBC was present in the soot, nor how much organic carbon;

3. we do not know the characteristics of the soot, which was from a hurricane lamp rather than a well-designed soot aerosol generator so spherule diameter and fractal dimension are unknown; and

4. as a consequence, we did not do the DMA→UHSAS tests.

With such poorly known aerosol, the results are hard to interpret, but it is clear that the UHSAS saw a similar overall concentration, so particles were not vaporizing fast enough to shrink below the detection limit. The largest particles were clearly undersized by the UHSAS, and the peak was undersized by about 10 %, representing 50 % less scattering than a PSL sphere with equivalent mobility diameter. That's actually much less undersized than the fullerene soot and Aquadag. The obvious interpretations are that either the kerosene flame actually produced little soot compared to organic matter or that the DLCA soot produced by the lamp scattered less, as you predicted, but did not heat up enough to vaporize, while the more compact Aquadag and fullerene particles absorbed much more light and vaporized. Denser, more compact particles both absorb more heat and diffuse it away less efficiently.

In addition, as you'll see in the new appendix, better scattering calculations do show that total scattering for compact aggregated absorbing spherules is significantly less than Mie calculations suggest. However, the side scatter seen by the UHSAS is actually enhanced. It doesn't cause a large sizing error because of the 6th power dependence of scattering on diameter. Absorption is also enhanced, which is directly related to particle heating.

**3. The calibration experiments are also missing a non-absorbing, non-spherical case, which would help to explain whether absorption is really important here, or just morphology. Silica or titania aggregates could be used (Schmoll et al., 2009) and would ideally be generated as DLCA aggregates (Eggersdorfer and Pratsinis, 2013) for comparison with soot. The authors may avoid this suggestion by including both soot and absorbing spheres, however. 4. In the scattering calculations corresponding to the calibrations, the authors should follow the literature to use the relatively simple RDGFA approach (Sorensen et al., 2018) rather than an effective medium approach in approximating soot properties. The combustion literature has long used the RDGFA approach to obtain reasonable results for soot and to show that soot scatters very little light at similar wavelengths (Liu et al., 2019). With the addition of an absorbing spherical particle type, a DLCA soot sample, and optionally a non-absorbing DLCA aggregate, the authors' work would represent a comprehensive study of the UHSAS response. So, these first 4 comments would not only close an important gap in the reasoning here but provide valuable reference data for others.**

[Figure]

Figure 1: Smoke from a kerosene lamp sampled simultaneously by the LDMA and the UHSAS. This is a 6 minute average. Yes, those are stitching errors at 0.13 and 0.24 µm.

Interesting ideas. As mentioned earlier, we have implemented better scattering calculations, but aren't in a position to run the recommended tests.

We do not have electron micrographs to verify this, but neither the fullerene soot nor the Aquadag used as surrogates for rBC are likely to resemble DLCA aggregates. They are generated in aqueous solution and then the water is evaporated in dried air then any organic materials are either evaporated or charred in the tube furnace. This treatment sort of resembles the processed rBC tested by Bambha and Michelsen (2015) that collapsed their DLCA particles into much more compact shapes.

**5. I have requested a direct demonstration of the evaporation hypothesis because I can propose an alternative hypothesis which the authors have not discussed: internal mixing of soot with non-absorbing or slightly absorbing material. Internal mixing is almost inevitable for plumes as old as those studied here (2 days to 2 weeks, Line 309). The authors' laboratory data shows that black-carbon surrogates scatter much less than predicted by equivalent spheres (as expected and noted in the previous comment).**

It appears we were not sufficiently clear here, as we certainly never even considered the possibility that particles we sampled in ORACLES were not internally mixed. Indeed, the SP2 data showed thick coatings were the norm. Fig. 11 in the paper showed that the fraction of rBC-containing particles roughly agreed with the fraction of anomalous particles, an indication (though not proof) that the anomalous particles were the ones containing rBC. The main source of our confusion was precisely that they ought to be a mixture that behaved differently than the nearly pure rBC particles tested in the lab.

You do bring to mind a possibility that we did not consider–that the non-anomalous particles also contained rBC, but in quantities too small for the SP2 to detect. Those could heat the particles enough to vaporize relatively volatile materials. We've added that possibility to the discussion.

**It can be expected that coated black carbon would behave somewhere in between soot and non-absorbing spheres, as suggested by previous work (Mikhailov et al., 2006). This does not univocally imply a continuous range of UHSAS signals in Figure 7, because particle breakup due to laser heating (Moteki and Kondo, 2007) could cause two UHSAS modes: either coatings evaporate and give a smaller signal, or they fragment and give a larger signal.**

In general, fragmentation would not increase scattering signal. If a single particle breaks into 8, for example, then the resulting particles will have half the diameter, but each will scatter $(1/2)^6 = 1/64$ as much, so total scattering drops by a factor of 8. (This is assuming particles smaller than $0.3\,\mu\text{m}$, so essentially within the Rayleigh scattering regime, and that all particles are in the beam simultaneously.) If they are absorbing particles, they would heat less in the beam so would not evaporate as quickly, but that seems unlikely to overwhelm the inverse relationship between number and scattering.

The Moteki and Kondo (2007) case posits a mixed rBC and non-absorbing OM particle where the two

parts split apart, so the non-absorbing part ceases to shrink and continues to scatter as it passes through the beam. If this happened in the UHSAS, the particle would still be undersized relative to the original particle diameter, but by much less than if the OM evaporated. It is certainly possible that this occurred during the inflight DMA→UHSAS tests, but if that was the typical cause of the undersizing of the $z = 1$ and $z = 2$ particles, we would have expected the SP2 to find a much higher fraction of particles containing rBC.

**The field data (authors' Figure 7) can therefore be explained simply as a mixture of soot, organic matter (OM), and soot+OM particles:**

**(a) The soot particles are the anomalous low scatterers with Dmob>100nm, as expected for DLCA aggregates.**

Possible, but the SP2 data indicates that most rBC had a thick coating. There does not appear to be a large population of uncoated rBC. It is also unlikely that our very aged soot particles still resemble DLCA aggregates. Electron micrographs in Miller et al. (2021) show much more compact shapes, appearing much more comparable to the processed rBC tested by Bambha and Michelsen (2015) with fractal dimension 2.3–2.4 rather than the 1.78 for DLCA aggregates in Sorensen (2011).

**(b) The OM particles are the smallest particles (circles and squares with Dmob<100nm).**
Sure.

**(c) The soot+OM particles are the larger particles (circles and squares with Dmob>100nm).**
But those particles are the most common, and the SP2 indicates that rBC-containing particles are generally only 5 to 18 % of the population. So this can only be the case if the rBC components are too small for the SP2 to detect.

**(d) The ratio of Dopt/Dmob (y axis of Figure 7) may decrease with increasing Dmob because larger soot particles have larger shape factors (Sorensen,2011), or because larger soot particles have larger internal coupling parameters (Sorensen et al., 2018), or both. I made a rough calculation of the latter effect and it appears to be smaller than the former.**

**A second internal-mixing hypothesis replaces soot with tarballs above.**
Yes, it may well be that tarballs behave very similarly to rBC.

**The current laboratory experiments on soot surrogates (Aquadag and fullerene soot) actually support this internal-mixing hypothesis more than the brown carbon evaporation hypothesis. These surrogates anyway probably scatter more light than soot since their structure is more compact, and since Aquadag is made up of larger graphite flakes. The authors may perhaps consider my alternative as a sub-set of their evaporation hypothesis, or disprove it using their thermal denuder data.**

It seems simpler to assume that there are only two major particle types: particles that contain rBC and thus absorb lots of light, heat up, and shrink considerably; and particles that are weakly absorbing, heat up less and lose only a relatively volatile fraction of their mass. The absorbing species in the latter particles could be BrC, tarballs, aged tarballs that are no longer recognizable as such, the intermediate BrC found by Adler et al. (2019), or even tiny amounts of rBC. This scheme agrees with the SP2 data indicating that the fraction of particles containing rBC is roughly comparable to the anomalous particle population, the thick coatings seen by the SP2, and the lab calibration of rBC surrogates.

**6. In the context of the previous comments, I question the value of a "correction" to the UHSAS. If the surprising signals represent real physical phenomena, and the UHSAS is working correctly, why "correct" the data? Section 3.2 could in- stead follow the tone of Section 3.1, and focus on the prediction of the UHSAS response from fundamental particle properties. Since the particle properties are not known exactly, the properties (morphology effects on scattering, morphology effects on Dmob, refractive index and – if justified – evaporated volume fraction) required to explain the observations can be discussed. If the authors' answer is that a correction is valuable to predict volumetric size distributions and total light scattering, then please modify the manuscript to emphasize this.**

To a large extent, I agree that a correction is futile–it utterly fails to account for the anomalously undersized particles and it's not likely to generalize to other projects.

However, the whole point of using the UHSAS in ORACLES was to rapidly measure particle size distributions through the entire range of important CCN diameters and to make detailed measurements of the optical properties of the aerosols. As it turns out, we got a lesson in the limitations of the UHSAS, and that

became the theme of this paper. Nevertheless, we'd like to make the UHSAS data as useful as possible. We hope it's clear that the correction is not universally useful.

**7. The manuscript has cited relevant SP2 work but there are a few points where the SP2 literature should be used to constrain the UHSAS predictions.**

**(a) The intensity of the SP2 laser has been reported as 1.7E+05 Wcm-2, 6.5E+05 W cm-2, and 4.05E+05 W cm-2 by Schwarz et al., 2006, Moteki and Kondo 2007, and Bambha and Michelsen 2015, respectively. Cai et al. (2008) reported the UHSAS laser intensity as 5.1E+05 W m-2. These are all similar to the 5.1E+09 W m-2 (5.1E+05 W cm-2) reported here. The SP2 and UHSAS wavelengths are about the same. So, the behaviour of a given particle in the SP2 can be extrapolated to the UHSAS. (While thinking about this I consulted Figure 1 of Corbin and Gysel-Beer 2019, which shows the behaviour of various particle types in the SP2 laser.) Any particles which vapourize (or carbonize) in the SP2 must vapourize in the UHSAS. Therefore, Equation 1 and Figure 4 cannot be correct. The SP2 routinely observes soot particles down to about 80 nm from their incandescence at >3000 K, yet Equation 1 apparently predicts only 1000 K at steady state for 100 nm soot. Moreover, this size is an overestimation for aggregate particles like fullerene soot. The assumptions behind Equation 1 must be flawed, at least for non-spherical particles. Bambha and Michelsen (2015) performed more detailed calculations than Cai et al. (2008); probably too detailed for this manuscript. So it becomes even more important to perform UHSAS calibration experiments with DLCA soot.**

Your argument is precisely correct. I hope we have sufficiently addressed it. (Except as noted above, while calibration with DLCA soot would be interesting, it is probably irrelevant for either the lab or the ORACLES aerosol.)

**(b) Sedlacek et al. (2018) showed that nigrosin (brown carbon / tarball surrogate) absorbs the SP2 laser. Corbin and Gysel-Beer (2019) reported SP2 time-resolve scattering cross-sections for the evaporation of tar brown carbon (TB) from heavy-fuel oil that was similar to Alexander et al. (2008)'s tarballs Their scattering cross-sections for TB actually look very similar to the "anomalous" particles reported here. In contrast, their scattering cross-sections for rBC actually show that the rBC evaporates before reaching the centre of the laser. So, how can the UHSAS see rBC? Presumably, the software uses the peak signal for all particles. This would correspond to the peak laser intensity for non-evaporating particles, but would occur before the peak for evaporating particles. This difference in incident laser intensity would result in an undersizing of evaporating particles, in addition to their actual change in volume.**

**(I have presumed that the software uses the peak signal because the SP2 would have to do the same if it did not have a "split" detector.) This is an important issue which will be implicitly addressed by the addition of soot particles in the calibrations.**

Yes, the UHSAS has peak detectors only, not the fast data acquisition that allows the SP2 to actually trace the peaks. Therefore, if we have a particle that is a mixture of rBC and more volatile material, scattering would be double-humped, as in Laborde et al. (2012) Fig. 1(c). But we have no way of knowing which peak the UHSAS detected. The same might well be true for tarballs, if coated with more volatile material. With BrC, presumably the heating would be less intense so there might only be one peak (unless the particles char and generate rBC, as nigrosin does in an SP2). So the odds are that any particle with significant absorption will be undersized, whether it is vaporizing sulfates, organic matter, tarballs, or rBC.

**Minor comments**

**While reading the manuscript carefully a number of minor comments arose, which I list here.**

**1. I appreciated the original structure of the introduction, but please add a short goals paragraph at the end.**

Okay

**Please also consider moving parts of Section 1.3 to Methods.**

It makes sense to move the absorption calculations to Methods.

**2. Please add an Appendix section where the prediction of nephelometer signals from**

UHSAS signals is explicitly described. I can imagine what was done but it should be spelt out.

Sure

**3. In Section 1.2 of the introduction, the authors' use of light-absorbing carbon (LAC) terminology could be refined. The section should cite Petzold et al. (2013) in its first paragraph, which is a review of the topics discussed there.**

I've changed the terminology to be consistent with Petzold et al. (2013).

**In the next paragraph, change boiling to incandescence (line 79) since rBC actually sublimates.**

I've changed the wording.

**Please reword the statement that BC is a hypothetical material that includes graphitic soot nanospheres and amorphous C, citing either the definitions given by Petzold et al. (2013), Bond and Bergstrom (2006). The word hypothetical suggests that soot is not of a consistent composition, which may mislead readers (Michelsen et al. 2020).**

I agree that "hypothetical" is misleading, as it implies that BC is not real. I meant to convey the ambiguity in the definition of BC (as is mentioned by Petzold et al. (2013)).

Michelsen et al. (2020) is a nice summary, but appears to be focused largely on soot derived from hydrocarbons, not biomass burning. "soot" is defined as "carbonaceous particles formed during the incomplete combustion or pyrolysis of hydrocarbons", neglecting the carbohydrates and other organic material that dominate biomass. I'm sure the processes generating soot from open biomass fires are largely similar to laboratory combustion of hydrocarbons, but there may be some added complexity.

**Please also reconsider the statement that quantitatively connecting the amount of carbon with the light absorption of LAC remains a challenge. The issue is not that connecting the two is challenging, but that the range of light-absorbing compounds formed by carbon is vast. These LAC compounds include BrC, black carbon, and amorphous carbon in tarballs (a recent summary is given in Corbin et al., 2019).**

As you say, the atmosphere has a plethora of varieties of LAC, generally all mixed together. Each has different optical characteristics that are not necessarily well known and are affected by the mixing state. Instruments that actually measure C cannot unambiguously differentiate between types of LAC. Instruments that measure optical properties cannot tell us how much C is present. On top of that is the difficulty that sampling techniques tend to alter the environment around the particles, changing their physical and chemical properties. Petzold et al. (2013) discuss the difficulty of reconciling chemical and optical measurements. There seem to be some challenges left!

**When the authors mention amorphous C, are they referring to the degree of graphitization (Michelsen et al., 2020) of the sample? The term amorphous C is often used for a specific carbon material, rather than amorphous domains within soot. Please reword to clarify.**

We do not know the precise chemical form of the ambient BC during ORACLES or even of the fullerene soot used in the lab tests. Michelsen et al. (2020) advocate a very restrictive definition of amorphous carbon that may not qualify as BC and is rather at odds with the usage of the term in older literature (e.g the description of fullerene soot in Gysel, Laborde, et al. 2011) that appears to include anything lacking long-range order, including glassy carbon, turbostratic carbon, polycrystalline carbon and perhaps carbon onions. Perhaps it will be useful in the future to refine the definitions that way.

**4. Please add a short description of the UHSAS to Methods. How are particle focussed into the beam, at what flow rate, etc.**

Done.

**5. Please mention the particle counter after the DMA in Methods. Of course, a DMA by itself does not provide size distributions, so I assume a CPC was used. Similarly, at line 126, please clarify that it is not the DMA but the CPC which cannot tell the difference between charges (especially important since this work uses a UHSAS to tell the difference).**

I've mentioned the CPCs and reworded the bit about doubly and triply charged particles somewhat.

**6. Line 147 mentions an empirical correction to a valve – please clarify if this correction was applied to CPC, UHSAS, or both. Please consider adding a graph to the supplement and/or stating here the magnitude of the correction.**

Clarified with the relevant equation and a slightly better explanation.

**7. Line 151 and surrounding, please mention what RH the sample would have had without desiccation or what the maximum dewpoint would have been (I assume it is very low given the altitude).**

It varies, but I've included the range.

**8. Line 161 please briefly mention the reason why the fraction is trivial, especially since the detector is missing from Figure 1. I am assuming that the reason is the small collection angle of the detector optics.**

I wish I could be quantitative here! I do not actually know the dimensions of either the avalanche photodiode or the PIN photodiode, as I have not opened up the optical chamber and I have found no mention of the sizes in the UHSAS documentation or in scientific literature. The photodiodes are roughly 5 cm from the particle beam. If I assume they are 1 cm in diameter (an overestimate, I suspect), then they subtend a solid angle of 0.031 sr, which is less than 1.2 % of the 2.65 sr collected by the Mangin optics.

**9. Figure 2 please change soot to ambient soot with a citation to Moteki et al (2010) in the legend. I originally misunderstood it as fullerene soot. Please also change graphite to graphite sphere (or similar) for clarity, since Aquadag is also graphite.**

Figure 2 has changed in response to another review. I think it is clearer.

**10. Line 180 please provide a citation for this description of fullerene soot. Please change to the RDG approximation here.**

The soot description is now referenced. As mentioned, I used MSTM rather than RGD.

**11. Line 191 "poor statistics"... how exactly was the statistical analysis done? Was the mode or median of the distributions used?**

Neither, actually. We used the particle by particle mode of the UHSAS, which reports the peak mV for each individual particle detected. We bin the values, plot a histogram, and report the most frequently occurring bin. I suppose that's akin to a mode. Description added.

**12. Line 192 please change "than expected" to "than predicted for equivalent spheres" or similar.**

Changed, but to "scatter far less light than the calculations suggest".

**13. Table 1, what is "amorphous C" here? I did not see this term used by Moteki et al. 2010. They did use the term "non-graphitic" to describe samples with a lower degree of graphitization. I suggest omitting this entirely as the optical properties of soot (one row above) will be similar. This comment relates to my general comment on LAC terminology above.**

It is the fullerene soot, which is $\geq 90$ % "amorphous carbon" according to Gysel, Laborde, et al. (2011). I think it is somewhat misleading to call it fullerene soot, because fullerenes are only a minor component and the major species controlling the optical properties ought to be emphasized. But I can see that it's clearer to use the name in common use.

**14. Table 1, please indicate either how these values were extrapolated from measurements at other wavelengths, or the measurement wavelength if they were not.**

We did no extrapolation except as already documented in the footnotes. All other values are either directly from the sources cited at wavelengths no more than a few tens of nanometers from 1054 or are interpolated from graphs in the sources cited. (Though it is not clear to me how Cai et al. (2008) determined $m$ for $NH_4NO_3$; the sources cited there are not particularly close to 1054 nm. In any case, we have no calibration data using $NH_4NO_3$ and no longer show it in the heating calculation figure, so it has been eliminated. )

**15. Table 1, please change 'variable' to the values used in order to convey more precise information.**

The whole point was that the effective refractive index changed with particle diameter. But, since I'm no longer using the Mie calculations for those materials, they are no longer in the table.

**16. Line 124, I do not agree that Gysel et al. (2011) concluded that Aquadag particles are 13% unknown composition after denuding at 450 degrees C, and I do not see the statement in that work. These particles would be 100% rBC by definition, but may have a different SP2 response to other forms of rBC. In other words, this statement is illogical since the SP2 is calibrated to the total mass of denuded particles. Perhaps the authors are referring here to**

the EC content of Aquadag; this excludes oxygen and other atoms, so is smaller than rBC mass.

Gysel, Laborde, et al. (2011) state that

> TC content accounts for ~83 % of the total gravimetrically determined mass, and consequently EC makes up ~76 % of the total mass. Tests with thermodenuding Aquadag particles at 400 °C before selecting them by mass resulted in ~15 % higher SP2 signal amplitude per particle mass compared to non-thermodenuded Aquadag.

So an undenuded 10 fg particle would have 7.6 fg of EC. A particle that passed through the denuder and had a mass of 10 fg would had 15 % more EC, or 8.7 fg. That's 13 % of the mass unaccounted for. It does not cause problems with SP2 calibrations because it's accounted for in the effective density calculation.

**17. Figure 6 should not show the extrapolation if the discussion describes it as "completely inappropriate", please harmonize.**

Changed.

**18. Line 250 please change "kernel function" to "transfer function" according to convention.**

"Kernel function" is hardly rare in the DMA literature (e.g. Gysel, McFiggans, et al. 2009; Talukdar and Swihart 2003), but that's usually from an inversion perspective. I'll admit that "transfer function" is easier to understand.

**19. Line 303 please report the wavelengths which the AAE was calculated from.**

Sure. It was 470 to 660 nm.

**20. Throughout the manuscript, error bars were generally missing and not discussed, please add them or an overall comment.**

Okay.

**21. Line 204 please mention 1/e2 as the measure of beam diameter (if correct).**

Okay.

**22. Line 382 much lower temperatures than 400 C, are enough to evaporate most coatings. e.g. https://doi.org/10.1016/j.jhazmat.2011.12.061**

Offhand, I'd agree that that's long been known. I cited Clarke (1991) for sulfates and Ellis and Novakov (1982) for organics vaporizing below 400 °C, but not there and I was not explicit. It's not actually obvious that Maruf Hossain et al. (2012) established that for ORACLES aerosol. The fuel and the aging were far different, and the ORACLES plume had modified combustion efficiencies (MCEs) $CO/(CO + CO_2) > 0.97$ (article in preparation), indicating efficient flaming combustion (Collier et al., 2016). Flaming combustion is where Maruf Hossain et al. (2012) found that their maximum temperature (250 °C) only evaporated 20% of the volume of 0.116 μm particles.

It is amusing to note that another reviewer cited Adler et al. (2019) to suggest exactly the opposite–that the coating might not be volatile at 400 °C. I hope the new discussion about this is clearer.

**References**

Adler, G., N. L. Wagner, K. D. Lamb, K. M. Manfred, J. P. Schwarz, A. Franchin, A. M. Middlebrook, R. A. Washenfelder, C. C. Womack, R. J. Yokelson, and D. M. Murphy (2019). "Evidence in biomass burning smoke for a light-absorbing aerosol with properties intermediate between brown and black carbon". *Aerosol Sci. Tech.* 53.9, pp. 976–989. DOI: 10.1080/02786826.2019.1617832.

Bambha, R. P. and H. A. Michelsen (2015). "Effects of aggregate morphology and size on laser-induced incandescence and scattering from black carbon (mature soot)". *J. Aerosol Sci.* 88, pp. 159–181. DOI: 10.1016/j.jaerosci.2015.06.006.

Cai, Y., D. C. Montague, W. Mooiweer-Bryan, and T. Deshler (2008). "Performance characteristics of the ultra high sensitivity aerosol spectrometer for particles between 55 and 800 nm: laboratory and field studies". *J. Aerosol Sci.* 39.9, pp. 759–769. DOI: 10.1016/j.jaerosci.2008.04.007.

Clarke, A. D. (1991). "A thermo-optic technique for in situ analysis of size-resolved aerosol physicochemistry". *Atmos Environ. A–Gen.* 25A.3/4, pp. 635–644. DOI: 10.1016/0960-1686(91)90061-B.

Collier, S., S. Zhou, T. B. Onasch, D. A. Jaffe, L. Kleinman, A. J. Sedlacek, N. L. Briggs, J. Hee, E. Fortner, J. E. Shilling, D. Worsnop, R. J. Yokelson, C. Parworth, X. Ge, J. Xu, Z. Butterfield, D. Chand, M. K. Dubey, M. S. Pekour, S. Springston, and Q. Zhang (2016). "Regional influence of aerosol emissions from wildfires driven by combustion efficiency: insights from the BBOP campaign". *Environ. Sci. Technol.* 50.16, pp. 8613–8622. DOI: 10.1021/acs.est.6b01617.

Ellis, E. C. and T. Novakov (1982). "Application of thermal analysis to the characterization of organic aerosol particles". In: *Atmospheric Pollution 1982*. Ed. by M. M. Benarie. Vol. 20. Studies in Environmental Science. Elsevier, pp. 227–238. DOI: doi.org/10.1016/S0166-1116(08)71009-4.

Gysel, M., M. Laborde, J. S. Olfert, R. Subramanian, and A. J. Gröhn (2011). "Effective density of Aquadag and fullerene soot black carbon reference materials used for SP2 calibration". *Atmos. Meas. Tech.* 4.12, pp. 2851–2858. DOI: 10.5194/amt-4-2851-2011.

Gysel, M., G. McFiggans, and H. Coe (2009). "Inversion of tandem differential mobility analyser (TDMA) measurements". *J. Aerosol Sci.* 40.2, pp. 134–151. DOI: https://doi.org/10.1016/j.jaerosci.2008.07.013.

Laborde, M., P. Mertes, P. Zieger, J. Dommen, U. Baltensperger, and M. Gysel (2012). "Sensitivity of the single particle soot photometer to different black carbon types". *Atmos. Meas. Tech.* 5.5, pp. 1031–1043. DOI: 10.5194/amt-5-1031-2012.

Mackowski, D. W. (2014). "A general superposition solution for electromagnetic scattering by multiple spherical domains of optically active media". *J. Quant. Spectrosc. Ra.* 133, pp. 264–270. DOI: https://doi.org/10.1016/j.jqsrt.2013.08.012.

Mackowski, D. W. and M. I. Mishchenko (2011). "A multiple sphere T-matrix Fortran code for use on parallel computer clusters". *J. Quant. Spectrosc. Ra.* 112.13, pp. 2182–2192. DOI: https://doi.org/10.1016/j.jqsrt.2011.02.019.

Mackowski, D. W. (2006). "A simplified model to predict the effects of aggregation on the absorption properties of soot particles". *J. Quant. Spectrosc. Ra.* 100.1. VIII Conference on Electromagnetic and Light Scattering by Nonspherical Particles, pp. 237–249. DOI: j.jqsrt.2005.11.041.

Maruf Hossain, A. M. M., S. Park, J.-S. Kim, and K. Park (2012). "Volatility and mixing states of ultrafine particles from biomass burning". *J. Hazard. Mater.* 205-206, pp. 189–197. DOI: https://doi.org/10.1016/j.jhazmat.2011.12.061.

McCoy, B. and C. Cha (1974). "Transport phenomena in the rarefied gas transition regime". *Chem. Eng. Sci.* 29.2, pp. 381–388. DOI: 10.1016/0009-2509(74)80047-3.

Michelsen, H. A., M. B. Colket, P.-E. Bengtsson, A. D'Anna, P. Desgroux, B. S. Haynes, J. H. Miller, G. J. Nathan, H. Pitsch, and H. Wang (2020). "A review of terminology used to describe soot formation and evolution under combustion and pyrolytic conditions". *ACS Nano* 14.10. PMID: 32986401, pp. 12470–12490. DOI: 10.1021/acsnano.0c06226.

Miller, R. M., G. M. McFarquhar, R. M. Rauber, J. R. O'Brien, S. Gupta, M. Segal-Rozenhaimer, A. N. Dobracki, A. J. Sedlacek, S. P. Burton, S. G. Howell, S. Freitag, and C. Dang (2021). "Observations of supermicron-sized aerosols originating from biomass burning in south central Africa". *Atmos. Chem. Phys. Disc.* 2021, pp. 1–23. DOI: 10.5194/acp-2021-414.

Moteki, N. and Y. Kondo (2007). "Effects of mixing state on black carbon measurements by laser-induced incandescence". *Aerosol Sci. Tech.* 41.4, pp. 398–417. DOI: 10.1080/02786820701199728.

Petzold, A., J. A. Ogren, M. Fiebig, P. Laj, S.-M. Li, U. Baltensperger, T. Holzer-Popp, S. Kinne, G. Pappalardo, N. Sugimoto, C. Wehrli, A. Wiedensohler, and X.-Y. Zhang (2013). "Recommendations for reporting "black carbon" measurements". *Atmos. Chem. Phys.* 13.16, pp. 8365–8379. DOI: 10.5194/acp-13-8365-2013.

Sorensen, C. M. (2011). "The mobility of fractal aggregates: a review". *Aerosol Sci. Tech.* 45.7, pp. 765–779. DOI: 10.1080/02786826.2011.560909.

Sorensen, C. M., J. Yon, F. Liu, J. Maughan, W. R. Heinson, and M. J. Berg (2018). "Light scattering and absorption by fractal aggregates including soot". *J. Quant. Spectrosc. Ra.* 217, pp. 459–473. DOI: https://doi.org/10.1016/j.jqsrt.2018.05.016.

Talukdar, S. S. and M. T. Swihart (2003). "An improved data inversion program for obtaining aerosol size distributions from scanning differential mobility analyzer data". *Aerosol Sci. Tech.* 37.2, pp. 145–161. DOI: 10.1080/02786820300952.

---

## Author Comment (AC3) · 7 Oct 2021

**Response to comments on "Undersizing of Aged African Biomass Burning Aerosol by an Ultra High Sensitivity Aerosol Spectrometer" by Richard Moore**

Steven Howell and Steffen Freitag

October 6, 2021

We'd like to thank Richard Moore for his review. It was particularly interesting to see his figure showing results similar to ours for Aquadag and fullerene soot, and his recommendation that calibrations be performed with partcles whose refractive index is close to that found in the deployment is wise. However, the pattern seen in the field calibration with aged smoke is clearly more dramatic than that seen in our data or that presented in the comment.

We contend that Mie scattering cannot explain our data and that heating in the laser is a plausible explanation. Dr. Moore's comment suggests that the real problem is that the UHSAS does not respond properly to Mie scattering and that refractive indices near 1.52 are simply sized poorly. (He does not propose that scattering from those particles might deviate significantly from Mie behavior.) For relatively large particles, where Mie wiggles hit, it is quite likely that given the uncertainties in refractive index, precise diameters, non-sphericity, and perhaps the actual (as opposed to specified) optical geometry of the UHSAS, the Mie calculations might be far off, as indeed they are in the lab calibrations. But for spherical particles with diameter much smaller than the wavelength, it is hard to see how the UHSAS could respond in a way that differs much from calculations. That leaves either particle geometry or changes in particle sizing due to heating as the likely causes.

**Comments from Richard Moore**

1. *The main peak undersizing of 0-15% (depending on size) is largely consistent with the difference in UHSAS size response between PSL particles and size-classified ammonium sulfate particles, and that the finding reported in the second bullet point above is due to differences in refractive index between the PSL calibration standard and aged, biomass burning aerosols.*

   But the pattern is different, with the BB particle undersizing much more strongly related to diameter. I see no indication that the $(NH_4)_2SO_4$ behavior deviates significantly from that predicted by Mie scattering (though it's hard to tell from a log-log plot). Note that when you plotted our data, the line actually crosses your 2020 data. As you'll see in the revised version of the paper, the volatility changes very little with diameter, suggesting that composition (and hence refractive index) is not a strong function of particle size. I'm also a bit curious what shape $(NH_4)_2SO_4$ particles have.

2. *Can the dual peaks between 100-180 nm be attributed to a stitching error in the transition region between the G3 to G2 gain stages, where the smaller 'anomalous' peak is from the G3 detector? While such a stitching error would not bias the polydisperse size distribution (or indeed even be noticeable in many cases), it may give rise to extra peaks when looking at monodisperse aerosols near the gain stage transition point. What was the G3 gain stage saturation diameter for this instrument during the ORACLES campaigns as well as the subsequent tests?*

   No, it's not a stitching error. Those tend to be very sharp and distinctive, while the anomalous peaks are of a breadth comparable to the other peaks (there is no smoothing applied in the figure). In addition, while stitching errors occur at fixed diameters (for a given calibration file) the anomalous particles show up at different diameters for different mobility diameters.

3. *Differences between Mie theory and the actual instrument performance may be substantial between 600-1000 nm as indicated by Figure 3, which may explain the poor performance of the extrapolated correction. The difference between NaCl (refractive index of 1.53+0i) and PSLs is particularly noticeable and seems to exhibit 20% undersizing between 600-1000 nm.*

   Yes, once the particles are large enough that Mie wiggles start to appear, theory and data seem to part company. The question is whether that is due to the particles not behaving like ideal Mie spheres or the instrument not responding to scattering as predicted. Certainly part of it is the former: particles aren't spheres and the refractive indexes aren't known perfectly. Some of it may be the latter: perhaps the optical angles aren't as well defined as hoped or the jet isn't through the center of the beam or there's a size-dependent defect in focusing the beam.

4. *Along the lines of my Point 1 and the authors' caution in the final bullet above, it's not clear to me that aged biomass burning particles or other atmospherically-relevant absorbing aerosols far from emissions sources are meaningfully different from non-absorbing aerosols in terms of UHSAS sizing. Instead, these results motivate the need to calibrate the UHSAS with particles of atmospherically- relevant refractive index instead of PSLs.*

   I emphatically agree, of course, since that's exactly what we were trying to do with the in-flight tests. The surprise was the extent of the undersizing, particularly for the anomalous particles. But for particles below 300 nm, much smaller than the 1054 nm laser, Mie calculations ought to be reliable!

   Part of the purpose of our lab work prior to deployment was to bracket the refractive indices we might see in the field, with higher (PSL) and lower ($H_2SO_4$) as well as something in the middle (NaCl).

*I'd suggest that the authors compare their biomass burning curve from Figure 6 to the NaCl curve from Figures 2-3 rather than relying only on Mie Theory calculations as in Figure 7 to rule out the refractive index explanation.*

As you requested, that is shown in Fig 1, including the airborne data from the plume and the NaCl data from the plume, along with the fit to the NaCl data using the same mathematical form, but extending the fit to 500 nm. The results are admittedly somewhat ambiguous, but the slope of the plume data is clearly steeper than the NaCl.

But it makes little sense to concentrate solely on the NaCl, when the other non-absorbing materials behaved quite differently. The $H_2SO_4$ is the only material we tested that is almost certainly spherical, and the UHSAS slightly oversized it until well into the Mie wiggles. The refractive index is not perfectly known, since it attracts water very effectively and also lab air typically has elevated $NO_3$, so there could have been a small fraction of $NH_3HSO_4$. Up to around 500 nm the $Na_2SO_4$ particles are very close (within 3%) to that predicted by the Mie calculations, even though the particles are not likely to be spherical and the refractive index used was for 589 nm, not the 1054 nm of the UHSAS laser.

The issue cannot simply be that the refractive index of NaCl is lower than that of PSL, since both sulfate particles have even lower refractive indices. The most obvious explanation is non-sphericity, even though it is true that the literature suggests it's not a really big factor. Perhaps the nebulizer we used in the lab or the drying rates were so different that the dynamic shape factor from Zieger et al. (2017) is optimistic and the resulting particles have significantly less mass. Of course if the mis-sizing of NaCl is due to non-sphericity, it suggests that perhaps the plume particles were strongly aspherical too. That seems unlikely for non-rBC particles since they are a product of vapor deposition to nucleation mode particles.

In any case, it would clearly be valuable to do a series of calibrations with definitely spherical particles having a variety of refractive indices to test whether the UHSAS does respond as predicted by Mie theory. It looks like you have been doing that, given the plot with numerous organics. Might my Mie code for scattering into UHSAS optics be of use?

*That the 'anomalous' particle size curve lines up so well with that for fullerene soot seems like more than a coincidence and would seem to imply the existence of a small, externally-mixed aerosol population in this aged biomass burning plume.*

Yes, it might, but that conflicts with the SP2 data, which showed thick coatings. Given the new heating calculations, it is inevitable that particles with even a small amount of rBC will get sufficiently hot to evaporate coatings, so the anomalous particles need not be made primarily of rBC.

[Figure]

Figure 1: UHSAS sizing of NaCl in the lab tests and of plume particles in flight. The solid blue curve is a fit of the same form as used for the smoke plume particles. The dashed blue line is the Mie calculation for spherical particles with the refractive index of NaCl after correcting for the DMA mis-sizing due to non-sphericity (Zieger et al. 2017).

**References**

Zieger, P., O. Väisänen, J. C. Corbin, D. G. Partridge, S. Bastelberger, M. Mousavi-Fard, B. Rosati, M. Gysel, U. K. Krieger, C. Leck, A. Nenes, I. Riipinen, A. Virtanen, and M. E. Salter (2017). "Revising the hygroscopicity of inorganic sea salt particles". 8.1, p. 15883. DOI: 10.1038/ncomms15883.